# ADAPT-FED: Adaptive Federated Optimization with Learning Stability

## Abstract

Federated Learning (FL) frequently exhibits poor generalization due to unstable training across heterogeneous clients. Although training instability can accelerate learning, it often compromises generalization, resulting in a fundamental tension within FL. This work introduces `ADAPT-FED`, a framework that adaptively regulates training dynamics to leverage the advantages of instability while mitigating its adverse effects. As a result, `ADAPT-FED` enables more stable and consistent learning in privacy-constrained environments. Experimental results on standard benchmarks demonstrate that `ADAPT-FED` enhances generalization and convergence relative to state-of-the-art FL optimization algorithms.

## 1 Introduction

Federated learning (FL) enables decentralized model training while preserving data privacy Li et al. (2019a); Wang et al. (2020b). However, FL implementation faces challenges due to the heterogeneity of clients' data distributions Hsieh et al. (2020), which complicates the aggregation of global model parameters, leading to poor generalization performance Li et al. (2019b).

Generalization is the model's ability to perform well on new, unseen data beyond the training dataset Zhang et al. (2021). In FL, robust generalization is essential for real-world applications where models face heterogeneous data and environmental conditions. Effective generalization prevents overfitting and guarantees the model's reliability across heterogeneous environments. Generalization is primarily pursued using first-order gradient methods (e.g., gradient descent (GD) and its variants Andrychowicz et al. (2016); Bottou (2010)) to minimize training loss during the learning process. However, challenges such as the absence of flat stationary points near the trajectory of first-order gradient methods Ahn et al. (2022), and differential privacy (DP) Dwork (2006) lead to training instability Abadi et al. (2016). This often results in non-monotonic reductions in the training loss as shown in Figure 1 (a), affecting the model's ability to generalize (models that train stably generalize well Chandramoorthy et al. (2022)). Interestingly, recent analyses suggest that such unstable convergence can sometimes accelerate

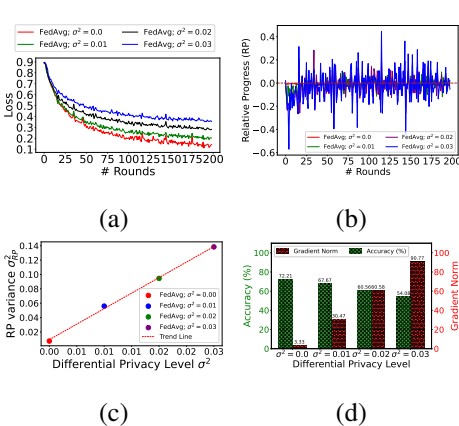

Figure 1: Correlation of DP noise with training instability based on CIFAR10 setup. The variance in relative progress shows that increased DP noise elevates instability, leading to larger gradient norms and lower accuracy.

optimization in centralized settings Ahn et al. (2022). This paradox motivates a critical but unexplored question in FL: *how can we adaptively regulate instability in FL to harness its optimization benefits while suppressing its negative effects on generalization?*

Recent developments in FL focus on improving generalization using sharpness-aware optimization. These techniques aim for flatter minima in the loss landscape. Adaptations such as FedSAM Qu et al. (2022) enhance generalization by applying sharpness-aware minimization at each client. This approach promotes local generalization. Adaptive optimization techniques Reddi et al. (2020) also aim to smooth the global loss surface, thereby improving generalization in FL. Despite these innovations, most methods address optimization only indirectly by seeking flatter minima or smoother updates. They do not directly control the training instability caused by the absence of flat stationary points

near the trajectory of first-order gradient methods and DP noise. The localized nature of current methods often cannot resolve the global stochasticity of FL environments. Thus, achieving robust global model performance when combining locally optimal updates remains challenging. There is a clear need for new approaches that go beyond flatness to refine local models and ensure stable, generalizable global learning—an essential but currently unmet need in FL.

To address the challenges above, we propose **Adapt**ive **Fed**erated Optimization with Learning Stability (`ADAPT-FED`), a framework designed to enhance both stability and generalization of FL models. `ADAPT-FED` dynamically adjusts learning rates based on historical relative progress (RP) metrics, which act as stability indicators within the optimization process. Specifically, `ADAPT-FED` increases the learning rate during stable periods and reduces it during unstable periods, ensuring consistent training progress and mitigating the typical instabilities caused by erratic updates. In designing and evaluating `ADAPT-FED`, we make the following contributions.

- We identify and analyze the causes of training instability and poor generalization in heterogeneous FL settings, focusing on the adverse effects of GD's lack of flat stationary points, partial client participation, and DP noise.
- We propose `ADAPT-FED`, the first FL framework that explicitly leverages the *relative progress (RP)* metric as an instability signal. Unlike prior adaptive or sharpness-aware approaches that either smooth updates or seek flatter minima, `ADAPT-FED` dynamically increases the learning rate when instability signals indicate faster progress is possible, and reduces it when instability threatens generalization.
- We theoretically validate the effectiveness of `ADAPT-FED` in mitigating training instability. Our analysis provides precise bounds on the improvements in stability and convergence rates, highlighting how `ADAPT-FED` mitigates the impact of training instability on the overall learning process.
- We conduct rigorous empirical evaluations demonstrating that `ADAPT-FED` significantly enhances model generalization and convergence across multiple datasets (CIFAR10, CIFAR100, and UTK), with improvements of up to $+\mathbf{5.06\%}$, $+\mathbf{14.79\%}$, and $+\mathbf{7.79\%}$ in generalization performance compared to SOTA FL algorithms.

## 2 RELATED WORK

**Sharpness-aware FL** focuses on adapting sharpness-aware optimization techniques Caldarola et al. (2022); Dai et al. (2023); Qu et al. (2022); Sun et al. (2023) to address the degradation of global model generalization under non-IID settings. Sharpness-aware optimization methods Cha et al. (2021); Izmailov et al. (2018); Foret et al. (2020); Kwon et al. (2021) improve generalization in centralized learning by seeking flatter minima in the loss landscape Foret et al. (2020); Kwon et al. (2021), which has inspired several adaptations for FL settings by prior work. For instance, FedSAM Qu et al. (2022) and its variants (FedGAMMA Dai et al. (2023), SWA Izmailov et al. (2018)) apply these optimizations locally at each client, promoting convergence to flatter local minima and improving local generalization. In conclusion, by minimizing loss and sharpness with smoother loss landscapes, sharpness-aware optimizations address client drift and improve both convergence and generalization across diverse and unseen data.

**Adaptive optimization techniques in FL.** focus on addressing the convergence challenges posed by heterogeneous client data and communication constraints. In particular, FedAdagrad Reddi et al. (2020) adjusts the learning rate based on the accumulated gradient squared values, making it effective for sparse-gradient tasks and ensuring that clients with less frequent updates still contribute meaningfully. FedAdam Reddi et al. (2020) builds on this by incorporating momentum terms to smooth out the optimization trajectory, offering robustness to noisy gradients. FedYogi Reddi et al. (2020) uses a more conservative update rule, reducing the risk of divergence in situations with large gradients. By adapting to the local landscape of each client, these optimizers ensure faster and more stable convergence, especially where simple methods like FedAvg McMahan et al. (2017) struggle due to the high variance in client updates.

**Limitations of existing techniques.** Despite recent innovations, most methods address optimization only indirectly. They seek flatter minima or smooth updates, but do not explicitly regulate the training instability. This instability is central to both convergence and generalization in FL. As a result, the localized focus of these approaches often fails to resolve the global stochasticity of FL. This leaves a persistent gap in achieving stable and reliable global learning.

## 3 PRELIMINARIES AND PROBLEM SETUP

To establish the context for our study, we define FL with DP and introduce the problem of *training instability* in FL. Our goal is to diagnose the role of instability in shaping convergence and generalization, thereby motivating the adaptive regulation strategy introduced later in this paper. The central question addressed through both theoretical and empirical analysis is: ***What is the effect of training instability on generalization in FL?*** By establishing this link, we motivate the need for methods that not only avoid instability but also regulate it adaptively.

### 3.1 COMMON FL AGGREGATION ALGORITHM (FEDAVG)

We consider a standard federated learning (FL) setting where a global model $\boldsymbol{\theta}$ is trained across $K$ decentralized clients. Each client $k$ holds local data $\mathcal{D}_k = \{(\boldsymbol{x}_k^n, y_k^n)\}_{n=1}^{N_k}$, which may differ in distribution across clients due to non-IID sampling and data distribution skew Hsieh et al. (2020); Liu et al. (2020). This *data heterogeneity* impairs convergence and generalization.

FL minimizes the global empirical risk:

$$\boldsymbol{\theta}^* \triangleq \arg\min_{\boldsymbol{\theta}} \left\{ F(\boldsymbol{\theta}^{t+1}) = \sum_{k=1}^K w_k F_k(\boldsymbol{\theta}_k^{t+1}) \right\}, \quad w_k = \frac{N_k}{\sum_j N_j}, \tag{1}$$

where $F_k$ is the local empirical risk on client $k$. Each client performs local training by initializing $\boldsymbol{\theta}_k^{t+1} \leftarrow \boldsymbol{\theta}^t$ and minimizing:

$$\boldsymbol{\theta}^* \leftarrow \arg\min_{\boldsymbol{\theta}} \left\{ (\boldsymbol{\theta} - \boldsymbol{\theta}_k^{t+1})^T \nabla F_k(\boldsymbol{\theta}_k^{t+1}) \right\} \text{ s.t. } ||\boldsymbol{\theta} - \boldsymbol{\theta}_k^{t+1}||^2 \le \epsilon. \tag{2}$$

The updated parameters are sent to the server and aggregated to form $\boldsymbol{\theta}^{t+1} = \sum_{k=1}^K w_k \boldsymbol{\theta}_k^{t+1}$.

### 3.2 PRELIMINARIES OF FLATNESS SEARCHING IN FL: FEDSAM

FedSAM leverages the SAM optimizer Foret et al. (2020) to enhance flatness exploration during local training in FedAvg Qu et al. (2022), aiming for more robust model performance.

**SAM optimizer:** The SAM optimizer transforms a loss function $F(\boldsymbol{\theta})$ into a min-max cost function as follows:

$$\min_{\boldsymbol{\theta}} \max_{||\delta|| \le \rho} F(\boldsymbol{\theta} + \delta), \tag{4}$$

where $\rho$ is a positive real number and $||\delta||$ is the L2-norm of $\delta$. As a key factor, $\delta$ works as the perturbation that maximally raises the loss value so that the SAM optimizer can find flat minima. The perturbation can be simply approximated as the gradient direction, which points to the steepest direction of the loss surface.

To preserve model privacy, FL applies *differential privacy (DP)* via Gaussian noise added to clipped local gradients. While DP prevents information leakage Shokri & Shmatikov (2015), it exacerbates *training instability* in FL. We formalize the DP mechanism and its sensitivity bounds in Appendix C.

### 3.3 INSTABILITY IN MACHINE LEARNING

Training instability in centralized learning Ahn et al. (2022) refers to the phenomenon in which GD in Equation 2 causes the local risk $F_k(\boldsymbol{\theta}_k^{t+1})$ to decrease *non-monotonically*. The instability occurs because GD trajectories infrequently encounter flat stationary points. Instead, the sharpness (curvature) $L$ fluctuates across training iterations Ahn et al. (2022). We give a rigorous analysis of instability in FL with SAM optimizer. Several key assumptions are outlined below (the proof appears in Appendix B).

**Assumption 1 (Lipschitz smoothness).** *The function $F_k$ is differentiable and $\nabla F_k$ is L-Lipschitz continuous, $\forall k \in \{1, 2, \ldots, K\}$, i.e.,*

$$||\nabla F_k(\boldsymbol{\theta}) - \nabla F_k(\boldsymbol{\theta}')|| \le L||\boldsymbol{\theta} - \boldsymbol{\theta}'||, \quad \forall \boldsymbol{\theta}, \boldsymbol{\theta}' \in \mathbb{R}^d. \tag{A.1}$$

**Assumption 2 (Bounded variance).** *The gradient of the function $F_k$ have $\sigma_l$-bounded variance, i.e.,*

$$\mathbb{E}_{\xi_e} \left[ ||\nabla F_k(\boldsymbol{\theta}^e(k); \xi_k) - \nabla F_k(\boldsymbol{\theta}(k))||^2 \right] \le \sigma_l^2, \quad \forall k \in \{1, 2, \ldots, K\}, e \in \{1, \ldots, E-1\}, \tag{A.2}$$

*and the global variance is also bounded, i.e.,*

$$\frac{1}{M} \sum_{k=1}^M ||\nabla F_k(\boldsymbol{\theta}) - \nabla F(\boldsymbol{\theta})||^2 \le \sigma_g^2, \quad \forall \boldsymbol{\theta} \in \mathbb{R}^d. \tag{A.3}$$

**Assumption 3 (Bounded gradient).** *For any $k \in \{1, 2, \ldots, K\}$ and $\boldsymbol{\theta} \in \mathbb{R}^d$, we have*

$$||\nabla F_k(\boldsymbol{\theta})|| \leq B. \tag{A.4}$$

**Theorem 3.1** (Expected Descent Bound). *Under Assumptions 1–3 (All variables and notation are defined in Appendix B), the expected loss after one communication round satisfies*

$$\mathbb{E}_{\mathcal{B}_k^e, \mathbf{z}_k^e}[F(\boldsymbol{\theta}^{t+1})]_{\mathcal{B}_k^e, \mathbf{z}_k^e} - \mathbb{E}[F(\boldsymbol{\theta}^t)] \leq \eta \pi^t E L^2(\rho^2 - B^2) - \frac{\pi^t \eta E}{2} ||\nabla F(\boldsymbol{\theta}^t)||^2 - \frac{\eta \pi^t}{2E} ||\frac{1}{K} \sum_{k=1}^{K} \Delta_k^t||^2$$

$$+ \frac{3\pi^t \eta^2 E}{2} \left( 2L^2 \rho^2 \sigma_i^2 + 6E(3\sigma_g^2 + 6L^2 \rho^2) + 6E||\nabla F(\boldsymbol{\theta}^t)||^2 \right) \quad (29)$$

$$+ \frac{3\pi^t \eta^2 E}{2} \left( 3\eta^2 E L^2 \rho^2 + B^2 \right) + \frac{3\eta^2 E L(L^2 \rho^2 + B^2)}{2}$$

$$+ \frac{L\sigma^2 C^2 pd}{2m^2}.$$

**Notation.** $\mathcal{B}_k^e$ is the stochastic mini-batch for client $k$ at iteration $e$. $\mathbf{z}_k^e$ is the DP noise added to the clipped gradient (e.g., Gaussian with variance $\sigma^2 C^2$).

The bound in Theorem 3.1 shows that sharp curvature terms ($L^2 \rho^2$) amplify instability and gradient fluctuations, while DP noise contributes an additional error floor ($\frac{L\sigma^2 C^2 pd}{2m^2}$). When curvature $L$ fluctuates due to lack of flat stationary points near GD trajectory Ahn et al. (2022), these effects compound, preventing consistent descent of the loss. This leads to unstable training dynamics and ultimately weakens generalization.

**Proposition 1** (Relative Progress as an Instability Metric (RP)). *Assume that Equation A.1 hold. We define RP:*

$$\text{RP}_t = \eta \cdot \underbrace{||g_k(\boldsymbol{\theta}_t) \cdot \min \left( 1, \frac{C}{||g_k(\boldsymbol{\theta}_t)||_2} \right) + \mathcal{N} \left( 0, \frac{\sigma^2 C^2}{m} \mathbf{I}_d \right) ||_2^2}_{\mathbb{E}_{\mathcal{B}_k^e, \mathbf{z}_k^e}[||\tilde{g}_k(\boldsymbol{\theta}_t)||_2^2]} \cdot \left( \mathbb{E}_{\mathcal{B}_k^e, \mathbf{z}_k^e}[F(\boldsymbol{\theta}_{t+1})] - F(\boldsymbol{\theta}_t) \right). \tag{3}$$

where $\eta$ is the learning rate, $C$ the gradient clipping constant, $\sigma^2$ the noise variance, $m$ the number of sampled clients, and $d$ the model dimensionality.

RP quantifies how much the global empirical risk improves after updating the gradients at each round relative to the size of the gradient and the step size $\eta$ taken. Stability in FL is achieved when the RP consistently remains below a negative threshold, indicating steady and controlled progress in the optimization process without erratic fluctuations.

**Proof of Proposition 1.** Assume that the global empirical risk $F(\boldsymbol{\theta})$ is $L$-smooth. Using the standard descent lemma for $L$-smooth functions and incorporating the DP-noised clipped gradient update, we have:

$$\mathbb{E}_{\mathcal{B}_k^e, \mathbf{z}_k^e}[F(\boldsymbol{\theta}^{t+1})] - F(\boldsymbol{\theta}^t) \leq -\eta \left( 1 - \frac{L\eta}{2} \right) \cdot \mathbb{E}_{\mathcal{B}_k^e, \mathbf{z}_k^e} \left[ ||\tilde{g}_k(\boldsymbol{\theta}_t)| \, |_2^2 \right], \tag{4}$$

$$\eta \, \mathbb{E}_{\mathcal{B}_k^e, \mathbf{z}_k^e} \left[ ||\tilde{g}_k(\boldsymbol{\theta}_t)||_2^2 \right] \left( \mathbb{E}_{\mathcal{B}_k^e, \mathbf{z}_k^e}[F(\boldsymbol{\theta}^{t+1})] - F(\boldsymbol{\theta}^t) \right) = -\eta^2 ||\nabla F(\boldsymbol{\theta})||^4 + \eta^3 ||\nabla F(\boldsymbol{\theta})||^2 \int_0^1 \tau \, \mathbb{E}_{\mathcal{B}_k^e, \mathbf{z}_k^e} \left[ ||g(\boldsymbol{\theta})||^2 L \right] d\tau$$

$$\leq -\eta^2 \left( 1 - \frac{L\eta}{2} \right) \mathbb{E}_{\mathcal{B}_k^e, \mathbf{z}_k^e} \left[ ||\tilde{g}_k(\boldsymbol{\theta}_t)||_2^3 \right]. \tag{5}$$

**Notation.** $\tau \in [0, 1]$ is the interpolation parameter used in the integral, which traces points along the line segment between the current iterate $\boldsymbol{\theta}$ and the update $\boldsymbol{\theta} - \eta g(\boldsymbol{\theta})$. It appears in the directional smoothness term $L(\boldsymbol{\theta}; \eta \tau g(\boldsymbol{\theta}))$ to capture the curvature information along this path.

*Takeaway: This formulation makes explicit how $RP$ reflects both the gradient magnitude and the local curvature through the expected Lipschitz constant $L$. As curvature increases, the second-order term grows linearly in $L$, offsetting the negative first-order descent term $-\eta^2 ||\nabla F(\boldsymbol{\theta})||^4$. The net effect is a less negative descent, i.e., reduced progress per step, which we capture as $\partial \text{RP}/\partial L > 0$. In this sense, curvature inhibits descent efficiency, making $RP$ positively associated with sharpness while descent efficiency $-\text{RP}$ decreases monotonically in $L$. Analyzing the descent inequality equation 5 reveals two regimes: 1) When $L < \frac{2}{\eta}$, the right-hand side (RHS) update term remains negative, ensuring each gradient step reduces the global empirical risk, promoting stable convergence. 2) Conversely, when $L > \frac{2}{\eta}$, the RHS term becomes positive, potentially increasing the global empirical risk at each step, leading to divergence and destabilizing the optimization.*

## 3.4 EMPIRICAL ANALYSIS OF INSTABILITY IN FL

As a preliminary study, we compute the instability $RP$ and generalization (accuracy) metrics of FedAvg for the CIFAR10 benchmark across FL rounds. We use the experimental setup in §D.2.

**Observation**: In Figure 1, the RP variance values across FL rounds are greater than zero, indicating training instability in FL. Higher levels of DP lead to increased RP variance, suggesting higher training instability. Additionally, higher gradient norm values, which are observed with increasing DP, signal slower convergence during training and a noticeable decline in generalization.

*Takeaway: In FL with DP, there exists an underlined optimal point in stability where the $\eta$ is optimized to maintain better generalization. Identifying the optimal $\eta$ enables maximization of both the model's convergence rate and its generalization capability by allowing all parameters to reach their optimal values, necessitating using larger $\eta$ for parameters that have a minimal impact on the model and smaller $\eta$ for those that significantly alter it.*

## 4 PROPOSED METHOD: ADAPT-FED

Following the preliminaries in Section 3, we describe the local training and aggregation steps of our method, ADAPT-FED. Our approach extends FedSAM, which stabilizes training through sharpness-aware perturbations that encourage flatter minima. Although FedSAM improves local generalization, it does not address instability from the lack of flat stationary points and DP noise. As a result, local perturbations may enhance sharpness but do not guarantee stable global convergence. To address this, ADAPT-FED introduces an adaptive mechanism that adjusts learning rates based on relative progress, providing stability-aware training that complements FedSAM's sharpness-aware updates.

### 4.1 TRAINING PROCESS OF ADAPT-FED

**Local training.** At the start of training round $t + 1$, client $k$ receives the aggregated global model $\boldsymbol{\theta}^t$ from the previous round $t$, initializes its local model with the global one $\boldsymbol{\theta}_k^{t+1} \longleftarrow \boldsymbol{\theta}^t$, and runs $E$ training epochs $\boldsymbol{\theta}_k^{t+1}$ with DP.

**Gradient Descent with DP.** Client $k$ trains $\boldsymbol{\theta}_k^{t+1}$ using GD to find the best local objective $F_k(\cdot)$ such that Equation 2 is satisfied. As GD progresses, the global model's training stability depends on the magnitude of the learning rate $\eta$ and the gradient norms $\mathbb{E}_{\mathcal{B}_k^e, \mathbf{z}_k^e}[\|g(\boldsymbol{\theta}^t)\|]$. When $\eta$ is chosen such that $L > \frac{2}{\eta}$, we have observed that the RHS term in Equation 5 becomes positive, which can increase the empirical risk at each step and lead to divergence, destabilizing the optimization process. To stabilize the optimization process, we must take into account a crucial piece of conventional wisdom originating from the quadratic Taylor approximation model of GD. According to this wisdom LeCun et al. (1992); Schaul et al. (2013), if the sharpness at local step $e$ is $L$, then $\eta$ should be set no larger than $\frac{2}{L}$ to prevent training instability. The $\eta = \frac{2}{L}$ rule continuously anneals the step size, ensuring that the training objective decreases at each iteration.

In practice, FedSAM extends GD by introducing sharpness-aware perturbations that effectively reduce the impact of high $L$, guiding updates toward flatter regions of the loss landscape.

**Challenges in Learning Rate Scheduling** Scheduling the learning rate using the $\eta = \frac{2}{L}$ rule results in small $\eta$ that hinder the learning process due to the progressive increase in $L$ at each training iteration, causing slow or even stalled convergence Cohen et al. (2021). This stalled convergence happens particularly when the model approaches areas of high sharpness (high sensitivity of the loss to perturbations in the parameter space) in the loss landscape, which are typically regions with steep gradients. Thus, the inverse relationship $\frac{2}{L}$ results in tiny $\eta$, potentially hindering convergence by making the steps too cautious and slow. It is also computationally expensive to compute $L$ at each iteration since it involves the second-order derivative of the objective function.

### 4.2 ADAPT-FED DYNAMIC LEARNING RATE ADJUSTMENT

To address the instability challenges that FedSAM alone cannot resolve, ADAPT-FED augments sharpness-aware training with an adaptive learning rate adjustment mechanism. Rather than relying solely on local perturbations, ADAPT-FED explicitly tracks relative progress (RP) as a signal of training stability and adjusts step sizes accordingly. This joint approach leverages the benefits of FedSAM's flatter minima while directly mitigating instability. We present the entire process of ADAPT-FED in Algorithm 1. Let $F(\boldsymbol{\theta})$ be an unstable objective function: a function differentiable w.r.t. parameters $\boldsymbol{\theta}$. We want to minimize the expected value of this function, $\mathbb{E}[F(\boldsymbol{\theta})]$, relative to its parameters, $\boldsymbol{\theta}$. For each client $k$, we use $\{RP_1^k, \ldots, RP_T^k\}$ to show the objective function's training stability measures at different FL training rounds $t \in \{1, \ldots, T\}$.

ADAPT-FED introduces a novel method for scheduling each client's local learning rate $\eta^k$. It dynamically schedules the $\eta^k$ based on the moving averages of the historical RP, where the hyperparameter $\beta > 0$ controls the decay rate of the moving average, allowing for precise control of GD steps based on the observed training instability. For each client $k$, ADAPT-FED calculates the moving average of RP values across training rounds $(\overline{RP}_t^k)$ to smooth out the measure of recent training progress over a window of $N$ iterations. This average is vital for assessing the overall direction and stability of the learning process $\overline{RP}_t^k = \frac{1}{N} \sum_{i=t-N+1}^{t} \exp(RP_i^k); \quad \forall i \in \{1, \ldots, t\}$. Inspired by LeCun et al. (1992); Schaul et al. (2013), which proposes that the $\eta^k$ should be chosen based on the inverse sharpness of the objective function $\eta_k = \frac{2}{L}$ that measures stability, ADAPT-FED schedules the $\eta^k$ for the next iteration based on the inverse of the moving average of $\overline{RP}_t^k$. This transformation, in

---

**Algorithm 1** `ADAPT-FED`: FedSAM with RP–adaptive learning rates

---

1: **Inputs:** clients $K$, rounds $T$, local epochs $E$, initial model $\boldsymbol{\theta}^0$, initial LR $\eta_0$, decay $\beta$, RP window $N$, SAM radius $\rho$, norm $p \in \{2, \infty\}$, DP clip $C$, noise scale $\sigma$, small $\varepsilon_{\text{num}}$

2: **Output:** global model $\boldsymbol{\theta}^T$

3: **for** $t = 1$ **to** $T$ **do**

4:     Server samples participating set $K$ of size $m$ and broadcasts $\boldsymbol{\theta}^t$

5:     **for all** clients $k \in K$ **in parallel do**

6:         $\text{RP}_t^k \leftarrow \eta^k \cdot \mathbb{E}_{\mathcal{B}_k^e, \mathbf{z}_k^e}[||\tilde{g}_k(\boldsymbol{\theta}_{t-1}^k)||_2^2] \cdot (\mathbb{E}_{\mathcal{B}_k^e, \mathbf{z}_k^e}[F(\boldsymbol{\theta}^{t-1})] - F(\boldsymbol{\theta}^{t-2}))$

7:         $\overline{\text{RP}}_t^k \leftarrow \frac{1}{N} \sum_{i=t-N+1}^{t} \exp(\text{RP}_i^k), \quad \eta^k \leftarrow \eta_0 \cdot \frac{\beta}{\overline{\text{RP}}_t^k}$           *(RP from prior round)*

8:         $\boldsymbol{\theta}_k^0 \leftarrow \boldsymbol{\theta}^t$

9:         **for** $e = 0$ **to** $E - 1$ **do**

10:           $g_k \leftarrow \nabla F_k(\boldsymbol{\theta}_k^e)$           *(Local training with FedSAM + DP)*

11:           $\delta_k \leftarrow \rho \cdot \dfrac{g_k}{||g_k||_p + \varepsilon_{\text{num}}}$           *(SAM perturbation)*

12:           $\tilde{\boldsymbol{\theta}}_k^e \leftarrow \boldsymbol{\theta}_k^e + \delta_k$

13:           $g_k^{\text{sam}} \leftarrow \nabla F_k(\tilde{\boldsymbol{\theta}}_k^e)$

14:           $\tilde{g}_k^{\text{dp}} \leftarrow g_k^{\text{sam}} \cdot \min\left(1, \frac{C}{||g_k^{\text{sam}}||_2}\right) + \mathcal{N}\left(0, \frac{\sigma^2 C^2}{m} I_d\right)$

15:           $\boldsymbol{\theta}_k^{e+1} \leftarrow \boldsymbol{\theta}_k^e - \eta_k^t \, \tilde{g}_k^{\text{dp}}$

16:         **end for**

17:         $\boldsymbol{\theta}_k^{t+1} \leftarrow \boldsymbol{\theta}_k^E;$   **send** $\boldsymbol{\theta}_k^{t+1}$ to server

18:     **end for**

19:     **Aggregate:** $\boldsymbol{\theta}^{t+1} \leftarrow \sum_{k \in S_t} w_k \, \boldsymbol{\theta}_k^{t+1}$           *(e.g., FedAvg weights)*

20: **end for**

---

which each $RP_i^k$ value is exponentiated before the moving average is calculated, has several benefits: 1) *The exponential function increases very rapidly, making it possible to assign more weight to higher RP values; thus, higher RP values will have a disproportionately larger learning rate $\eta^k$ scheduling effect for enhanced stability.* 2) *If the RP includes negative values, the exponential function ensures all transformed RPs are positive to guarantee positive learning rates.* This scaling is designed to stabilize the training dynamically, responding to the immediate past training stability conditions $\eta^k = \eta_0 \cdot \left(\frac{\beta}{RP_t^k}\right)$.

**Intuition:** `ADAPT-FED` fine-tunes $\eta^k$ to match the actual training dynamics. When the $RP$ is low, indicative of stable progress, $\eta^k$ increases, which is conducive to faster convergence. Conversely, high $RP$ signals training instability, prompting a reduction in the $\eta^k$ to safeguard against potential divergences, mitigating training instability.

Based on the learning rate scheduling procedure, we perform the local model update as $\boldsymbol{\theta}_k^{e+1} = \boldsymbol{\theta}_k^e - \eta_0 \cdot \left(\frac{\beta}{RP_t^k}\right) \cdot \nabla F_k(\boldsymbol{\theta}_k^e)$. Each training round $t$ ends with the termination of local training and the return of updated local models to the server for aggregation into a global model.

**Server Aggregation:** The updated local models are then aggregated at the server to newly update the global model $\boldsymbol{\theta}^{t+1}$ for the next round. We adopt the commonly used FedAvg aggregation scheme to aggregate local models into a global model $\boldsymbol{\theta}^{t+1} = \sum_{k=1}^{K} w_k \boldsymbol{\theta}_k^{e+1}$.

### 4.3 ADAPTIVE LEARNING RATE COMPONENTS

This section outlines methods for setting the learning rate decay constant $\beta$ and the initial learning rate $\eta_0$.

#### 4.3.1 LEARNING RATE DECAY CONSTANT $\beta$

Choose $\beta$ to adapt $\eta^k$ responsively across rounds. Motivated by edge-of-stability theory Cohen et al. (2021) (and the classical $2/L$ stability threshold), set $\beta = 2$ so that $\eta^k = \eta_0 \cdot \left(\frac{2}{RP_t^k}\right)$, which makes $\eta^k$ sensitive to

instability magnitude. *As $\overline{RP_t^k}$ decreases in sharper regions, the step size contracts, helping cap fairness drift when RP spikes.*

### 4.3.2 INITIAL LEARNING RATE $\eta_0$

We select $\eta_0$ via a learning-rate range test Smith (2017), scanning a practical range and choosing the region with best generalization. *Anchoring $\eta_0$ well improves both convergence and fairness by keeping subsequent RP-driven adjustments within a stable, low-drift regime.*

## 5 THEORETICAL ANALYSIS

This section discusses the theoretical bounds of ADAPT-FED, focusing on its convergence rate. We provide theorems that set upper bounds on how quickly ADAPT-FED can stably converge, leading to performance generalization. These theorems are essential for understanding how the *eta* affects convergence. Before that, we introduce the assumptions consistent with other works in FL Li et al. (2019b):

### 5.1 CONVERGENCE ANALYSIS OF ADAPT-FED

**Theorem 5.1** (Expected Descent with ADAPT-FED Scheduling). *Let Assumptions 1–3 hold (All variables and notation are defined in Appendix B). In round t, each client k uses the adaptive local learning rate*

$$\eta_k^t = \eta_0 \cdot \frac{\beta}{RP_t^k}, \qquad \bar{\eta}_t = \frac{1}{K} \sum_{k \in K} \eta_k^t \quad (mean \quad step-size).$$

$$
\begin{aligned}
\mathbb{E}_{\mathcal{B}_k^e, \mathbf{z}_k^e}[F(\boldsymbol{\theta}^{t+1})] - F(\boldsymbol{\theta}^t) \leq & -\frac{\pi^t(\frac{1}{K}\sum_{k \in K}\frac{\eta_0\beta}{RP_t^k})E}{2}||\nabla F(\boldsymbol{\theta}^t)||^2 + \frac{(\frac{1}{K}\sum_{k \in K}\frac{\eta_0\beta}{RP_t^k})^2 EL^2\rho^2}{2} \\
& + \frac{3\pi^t(\frac{1}{K}\sum_{k \in K}\frac{\eta_0\beta}{RP_t^k})^2 EL^2\rho^2}{2} + \frac{15\pi^t E(\frac{1}{K}\sum_{k \in K}\frac{\beta}{\eta_0 RP_t^k})^2 L^2}{2} \\
& + \frac{3\pi^t(\frac{1}{K}\sum_{k \in K}\frac{\eta_0\beta}{RP_t^k})^2 EL(L^2\rho^2 + B^2)}{2} + \frac{L\sigma^2 C^2 pd}{2m^2}.
\end{aligned}
\tag{30}
$$

*Takeaway: ADAPT-FED enables self-regulating optimization. Client-wise RP directly controls both progress and error terms, resulting in larger, safer steps in flat regions and smaller, protective steps in sharp or noisy regimes. This approach reduces loss fluctuations and enhances.*

## 6 EXPERIMENTS

We extensively evaluate ADAPT-FED's effectiveness in achieving generalization for FL with DP under different data heterogeneity levels while adhering to two constraints: maintaining performance stability; and faster convergence.

### 6.1 EXPERIMENTAL SETUP

**Models and datasets.** We assess ADAPT-FED's efficacy using the setup in §D.2. We compare ADAPT-FED with SOTA baselines on the FL classification benchmarks datasets CIFAR10, CIFAR10, and UTK, examining generalization across different client partitions in FL.

**Baselines:** We evaluate ADAPT-FED across three key categories: 1) *FL baseline category* represented by FedAvg, serves as the standard learning scheme in FL. 2) *FL sharpness-aware category* includes FedSAM and FedASAM Caldarola et al. (2022); Dai et al. (2023); Qu et al. (2022); Sun et al. (2023), which flattens minima in the loss landscape to improve model generalization. 3) *FL regularization category* includes FedProxMohri et al. (2019), which uses regularization techniques to minimize the divergence of local models for improved model generalization.

**Hyperparameters.** For each case of algorithm and its evaluation on the benchmarks, we tuned the hyperparameters: $\mu$ for FedProx is tuned among three choices $\{0.01, 0.1, 1\}$. We tuned the hyperparameters $\rho$ of FedSAM and FedASAM among three choices $\{0.02, 0.05, 0.1\}$, and their respective $\beta \in \{0.1, 0.9\}$. Finally, we set the initial local learning rate using grid search as $\eta_0 = \{0.09, 0.04, 0.1, 0.3\}$. We set the noniid-ness $\alpha = \{0.3, 0.05\}$, DP $\sigma^2 = \{0.0, 0.01, 0.02, 0.03\}$, and DP clipping constant $C = 1$ for all the evaluations in the main paper. We present empirical results across both 10 and 20 clients. Detailed ablation studies for these hyperparameters and their impact on model generalization and convergence speed are reported in **??**.

### 6.2 PERFORMANCE EVALUATION

#### 6.2.1 GENERALIZATION ANALYSIS FL

ADAPT-FED outperforms SOTA techniques in heterogeneous FL settings as shown in Table 3 (refer to Appendix E.1 for detailed generalization evaluations). ADAPT-FED demonstrates generalization improvements of up to +5.06%, +14.79%, and +7.79% for CIFAR10, CIFAR100, and UTK respectively. These results confirm that ADAPT-FED effectively mitigates the strong training instability associated with heterogeneity, thereby enhancing generalization across clients. We believe these generalization gains are largely due to ADAPT-FED's use of stability-based adaptive learning rates, which directly address training instabilities caused by the GD learning algorithm. In contrast, existing techniques for training stability primarily focus on instabilities caused by discrepancies in local models due to data heterogeneity across clients, which does not inherently guarantee stability in GD learning. As heterogeneity is alleviated, as $\alpha$ increases from 0.05 to 0.3, generalization performance across all baselines improves due to the homogeneity of data distribution, which reduces discrepancies between local models across clients. Nevertheless, ADAPT-FED continues to demonstrate superior capability in enhancing generalization.

Table 1: $\lambda_{\max}$ results for CIFAR10.

| Algorithm | IID ($\epsilon = 0.53$) |
|---|---|
| | $\lambda_{\max}$ ↓ |
| FedAvg | $92.23 \pm 1.05$ |
| FedSAM | $23.25 \pm 0.62$ |
| FedASAM | $23.51 \pm 0.58$ |
| FedProx | $94.47 \pm 1.12$ |
| FedAdagrad | $24.05 \pm 0.71$ |
| FedAdam | $34.10 \pm 0.84$ |
| FedYogi | $91.90 \pm 1.20$ |
| ADAPT-FED (ours) | $\mathbf{18.90 \pm 0.44}$ |

↓: lower is better.

***Takeaway:*** *ADAPT-FED improves generalization compared to SOTA techniques in non-IID FL environments.*

Table 2: Generalization performance of ADAPT-FED versus baseline algorithms across 10 clients on three datasets: CIFAR10, CIFAR100, and UTK ($\eta_0 = 0.1$). For readability, only the mean values across 3 runs are shown. Best means are in bold.

| Algorithm | CIFAR-10 | | | | | | | | CIFAR-100 | | | | | | | | UTK | | | | | | | |
|---|---|---|---|---|---|---|---|---|---|---|---|---|---|---|---|---|---|---|---|---|---|---|---|---|
| | Dir. ($\alpha = 0.05$, non-IID) | | | | Dir. ($\alpha = 0.3$) | | | | Dir. ($\alpha = 0.05$, non-IID) | | | | Dir. ($\alpha = 0.3$) | | | | Dir. ($\alpha = 0.05$, non-IID) | | | | Dir. ($\alpha = 0.3$) | | | |
| | $\epsilon = +\infty$ | 1.85 | 0.53 | 0.23 | $+\infty$ | 1.85 | 0.53 | 0.23 | $+\infty$ | 1.85 | 0.53 | 0.23 | $+\infty$ | 1.85 | 0.53 | 0.23 | $+\infty$ | 1.85 | 0.53 | 0.23 | $+\infty$ | 1.85 | 0.53 | 0.23 |
| FedAvg | 58.76 | 53.72 | 47.31 | 41.91 | 74.55 | 70.18 | 64.76 | 59.68 | 37.73 | 37.03 | 33.94 | 30.89 | 41.21 | 40.91 | 37.48 | 34.84 | 64.11 | 64.11 | 55.18 | 54.98 | 78.89 | 78.89 | 64.87 | 64.83 |
| FedSAM | 57.93 | 58.08 | 58.67 | 58.74 | 74.80 | 74.18 | 74.61 | 74.90 | 37.43 | 38.49 | 38.57 | 38.17 | 42.18 | 42.64 | 42.81 | 43.89 | 73.28 | 73.28 | 73.58 | 73.35 | 79.06 | 79.06 | 79.10 | 78.83 |
| FedASAM | 59.09 | 58.78 | 58.74 | 58.74 | 75.40 | 74.91 | 74.51 | 74.51 | 38.56 | 37.83 | 37.99 | 37.99 | 43.39 | 43.12 | 43.28 | 43.28 | 73.31 | 73.31 | 74.07 | 74.07 | 78.63 | 78.63 | 79.48 | 79.48 |
| FedProx | 59.86 | 55.27 | 49.14 | 42.82 | 73.54 | 69.48 | 64.36 | 59.91 | 37.90 | 36.37 | 34.30 | 30.97 | 42.36 | 40.66 | 36.87 | 34.84 | 65.42 | 65.42 | 55.05 | 54.68 | 71.40 | 71.40 | 57.03 | 56.65 |
| FedAdagrad | 58.76 | 53.72 | 47.31 | 41.91 | 74.55 | 70.18 | 64.76 | 59.68 | 37.73 | 37.03 | 33.94 | 30.89 | 41.21 | 40.91 | 37.48 | 34.84 | 64.11 | 64.11 | 55.18 | 54.98 | 71.06 | 71.06 | 56.38 | 56.56 |
| FedAdam | 58.76 | 53.72 | 47.31 | 41.91 | 74.55 | 70.18 | 64.76 | 59.68 | 37.73 | 37.03 | 33.94 | 30.89 | 41.21 | 40.91 | 37.48 | 34.84 | 64.11 | 64.11 | 55.18 | 54.98 | 78.89 | 78.89 | 64.87 | 64.83 |
| FedYogi | 58.76 | 53.72 | 47.31 | 41.91 | 74.55 | 70.18 | 64.76 | 59.68 | 37.73 | 37.03 | 33.94 | 30.89 | 41.21 | 40.91 | 37.48 | 34.84 | 64.11 | 64.11 | 55.18 | 54.98 | 78.89 | 78.89 | 64.87 | 64.83 |
| ADAPT-FED (ours) | **63.02** | **65.18** | **65.39** | **65.83** | **80.46** | **81.33** | **81.24** | **81.75** | **51.44** | **53.59** | **54.26** | **54.34** | **58.18** | **61.25** | **61.38** | **60.39** | **75.05** | **75.05** | **75.05** | **74.31** | **86.85** | **86.85** | **86.49** | **86.86** |

Table 3: Generalization performance of ADAPT-FED versus baseline algorithms across 20 clients on three datasets: CIFAR10, CIFAR100, and UTK ($\eta_0 = 0.1$). For readability, only the mean values across 3 runs are shown. Best means are in bold.

| Algorithm | CIFAR-10 | | | | | | | | CIFAR-100 | | | | | | | | UTK | | | | | | | |
|---|---|---|---|---|---|---|---|---|---|---|---|---|---|---|---|---|---|---|---|---|---|---|---|---|
| | Dir. ($\alpha = 0.05$, non-IID) | | | | Dir. ($\alpha = 0.3$) | | | | Dir. ($\alpha = 0.05$, non-IID) | | | | Dir. ($\alpha = 0.3$) | | | | Dir. ($\alpha = 0.05$, non-IID) | | | | Dir. ($\alpha = 0.3$) | | | |
| | $\epsilon = +\infty$ | 1.85 | 0.53 | 0.23 | $+\infty$ | 1.85 | 0.53 | 0.23 | $+\infty$ | 1.85 | 0.53 | 0.23 | $+\infty$ | 1.85 | 0.53 | 0.23 | $+\infty$ | 1.85 | 0.53 | 0.23 | $+\infty$ | 1.85 | 0.53 | 0.23 |
| FedAvg | 41.83 | 41.83 | 36.20 | 32.72 | 62.29 | 59.05 | 53.79 | 50.27 | 22.12 | 21.77 | 21.15 | 19.74 | 41.81 | 41.24 | 37.66 | 34.20 | 61.71 | 61.71 | 53.73 | 53.94 | 74.28 | 74.28 | 60.14 | 60.20 |
| FedSAM | 42.45 | 44.40 | 45.02 | 45.02 | 60.85 | 60.41 | 60.73 | 59.00 | 18.23 | 20.17 | 20.53 | 19.53 | 41.52 | 42.32 | 41.90 | 42.96 | 72.98 | 72.98 | 72.77 | 72.13 | 82.22 | 82.22 | 82.78 | 82.43 |
| FedASAM | 45.57 | 45.49 | 45.89 | 45.89 | 61.91 | 61.49 | 61.07 | 61.07 | 22.29 | 22.57 | 21.91 | 21.91 | 42.82 | 42.19 | 42.60 | 42.60 | 72.19 | 72.19 | 73.25 | 73.25 | 83.30 | 83.30 | 82.68 | 82.68 |
| FedProx | 46.90 | 42.64 | 37.10 | 33.85 | 61.50 | 57.51 | 54.37 | 50.03 | 21.93 | 21.98 | 21.15 | 20.12 | 42.10 | 41.36 | 37.62 | 34.15 | 61.70 | 61.70 | 53.64 | 53.26 | 74.01 | 74.01 | 60.93 | 59.08 |
| FedAdagrad | 45.24 | 41.83 | 36.20 | 32.74 | 62.30 | 59.10 | 53.79 | 50.28 | 22.13 | 21.62 | 21.15 | 19.74 | 41.83 | 41.35 | 37.77 | 34.50 | 61.89 | 61.89 | 53.73 | 53.95 | 74.30 | 74.30 | 60.41 | 60.43 |
| FedAdam | 45.24 | 41.83 | 36.20 | 32.74 | 62.29 | 59.05 | 53.91 | 50.47 | 22.13 | 21.62 | 21.15 | 19.74 | 41.78 | 41.56 | 37.61 | 34.25 | 61.89 | 61.89 | 53.73 | 53.95 | 74.28 | 74.28 | 60.14 | 60.20 |
| FedYogi | 45.24 | 41.83 | 36.20 | 32.74 | 62.47 | 59.44 | 53.27 | 50.14 | 22.13 | 21.62 | 21.15 | 19.74 | 41.82 | 41.56 | 37.69 | 34.20 | 61.89 | 61.89 | 53.73 | 53.95 | 74.91 | 74.48 | 60.27 | 60.20 |
| ADAPT-FED (ours) | 43.39 | **49.99** | **50.78** | **49.55** | **72.01** | **73.88** | **73.88** | **74.60** | 7.75 | **23.08** | 13.12 | **24.29** | **48.41** | **52.48** | **54.88** | **56.13** | **79.20** | **79.20** | **76.22** | **76.22** | **84.95** | **84.95** | **85.13** | **84.13** |

### 6.2.2 RATE OF CONVERGENCE ANALYSIS

We compare ADAPT-FED with SOTA techniques to evaluate its ability to achieve faster convergence. On the CIFAR10, CIFAR100, and UTK datasets, ADAPT-FED demonstrates faster and more robust convergence than the baselines as shown in Figure 2. The improved convergence rate is a direct result of ADAPT-FED's use of adaptive learning rates, which specifically address training instabilities caused by the GD learning algorithm. In contrast, other techniques mainly focus on mitigating instability arising from discrepancies in local models due to data heterogeneity across clients, which does not inherently ensure stable learning.

***Takeaway:*** *ADAPT-FED leads to faster and more robust convergence compared to SOTA techniques in FL with DP (For additional experiments, including comprehensive generalization comparisons, convergence behavior under heterogeneity, and training stability across privacy levels, refer to Appendix E).*

### 6.2.3 PRIVACY-UTILITY EVALUATION

Table 3 presents the test accuracies corresponding to different levels of privacy guarantees. ADAPT-FED consistently surpasses previous state-of-the-art methods across a range of privacy budgets $\epsilon$. The enhanced convergence rate is attributed to ADAPT-FED's implementation of adaptive learning rates, which mitigate training instabilities introduced by the GD under differential privacy noise.

Table 4: RP variation and generalization performance across three datasets. Top: results under different $\eta_0$ values. Bottom: results under different $N$ values.

| | Variation with $\eta_0$ | | | | | |
|---|---|---|---|---|---|---|
| | CIFAR10 | | CIFAR100 | | UTK | |
| | $\eta_0 = 0.04$ | $\eta_0 = 0.1$ | $\eta_0 = 0.04$ | $\eta_0 = 0.1$ | $\eta_0 = 0.04$ | $\eta_0 = 0.1$ |
| RP Variation | 1.98 | 2.10 | 1.66 | 1.83 | 2.23 | 2.31 |
| Generalization | $77.48 \pm 0.42$ | $73.88 \pm 0.38$ | $57.40 \pm 0.35$ | $54.88 \pm 0.33$ | $85.56 \pm 0.28$ | $84.95 \pm 0.27$ |
| | Variation with $N$ | | | | | |
| | CIFAR10 | | CIFAR100 | | UTK | |
| | $N = 1$ | $N = N_{max}$ | $N = 1$ | $N = N_{max}$ | $N = 1$ | $N = N_{max}$ |
| RP Variation | 2.01 | 2.10 | 1.71 | 1.83 | 2.12 | 2.31 |
| Generalization | $75.63 \pm 0.40$ | $73.88 \pm 0.38$ | $56.34 \pm 0.34$ | $54.88 \pm 0.33$ | $86.10 \pm 0.27$ | $84.95 \pm 0.27$ |

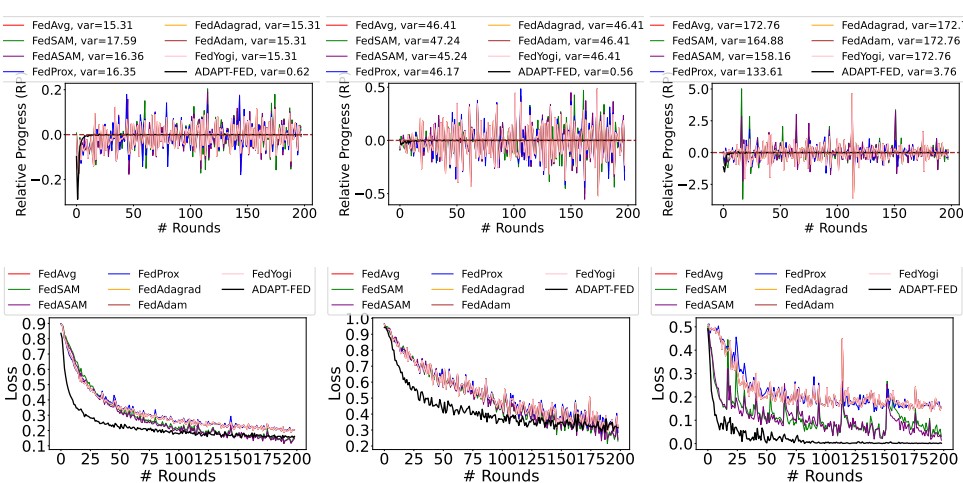

Figure 2: Convergence of the training loss of `ADAPT-FED` and baseline algorithms on 10 clients (CIFAR10, CIFAR100, and UTK, noniid-ness $\alpha = 0.3$) with DP $\sigma^2 = 0.01$, $\eta_o = 0.1$.

### 6.2.4 FLATNESS RESULTS

To evaluate the flatness of the global model, we report the maximum eigenvalue of the Hessian, $\lambda_{\max}$, on CIFAR10, a standard proxy for sharpness Qu et al. (2022). A lower value indicates flatter minima and better stability. While baselines either converge to sharp regions (e.g., FedAvg, FedProx, FedYogi) or partially reduce sharpness through perturbation-based methods (FedSAM, FedASAM), `ADAPT-FED` consistently achieves the lowest $\lambda_{\max}$. This result highlights that adaptive learning rate regulation guided by relative progress is more effective than static sharpness-aware updates, producing flatter solutions and mitigating DP-induced instabilities. These flatter minima explain the faster and more stable convergence observed in `ADAPT-FED`.

### 6.2.5 ABLATION STUDY

We study the effect of two key components in `ADAPT-FED`: the initial learning rate $\eta_0$ and the RP smoothing window $N$. Table 4 shows that increasing $\eta_0$ from 0.04 to 0.1 consistently raises RP variation across all datasets, leading to reduced generalization. This confirms that larger step sizes amplify instability, which degrades performance even when adaptive scheduling is applied. Similarly, Table 4 compares short vs. long RP smoothing. Larger $N$ increases RP variation and lowers accuracy, suggesting that excessive smoothing makes the controller less responsive to instability spikes. stability-aware learning requires careful calibration: overly aggressive $\eta_0$ or excessive smoothing undermines the ability of `ADAPT-FED` to regulate instability, while moderate values strike the best balance between stability and generalization.

## 7 CONCLUSION

We introduce `ADAPT-FED`, an FL method that tackles the challenges of training instability and suboptimal generalization in FL. `ADAPT-FED` dynamically adjusts learning rates based on historical relative progress metrics, enhancing stability and improving the generalization across clients with heterogeneous data. We establish a detailed theoretical framework analyzing how `ADAPT-FED` mitigates the impacts of data heterogeneity and gradient noise on the learning process. Our theoretical findings are supported by empirical evaluations across various datasets, where `ADAPT-FED` consistently outperforms SOTA optimization methods in improving stability, accelerating convergence, and generalization. These improvements make `ADAPT-FED` a robust solution for practical, real-world FL applications.

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

# Appendix
We provide additional information for our paper, `ADAPT-FED: Adaptive Federated Optimization with Learning Stability`, in the following order:

- Preliminary Lemmas (Appendix A)
- Proof of Convergence Analysis (Appendix B)
- Differential Privacy Guarantees in Federated Learning (Appendix C)
- Detailed Preliminaries and Problem Setup (Appendix D)
- Additional Experimental Results (Appendix E)
- Alternative RP-Based $\beta$ Scheduling Mechanisms (Appendix F)

## A  PRELIMINARY LEMMAS

**Lemma 1 (Lemma B.1, Qu et al. (2022)).** Under Assumptions 1–2, for any learning rate $\eta \leq \frac{1}{4EL}$, the updates exhibit drift arising from the deviation $\delta_{k,e} - \delta$.

$$\frac{1}{K} \sum_k \mathbb{E}\big[||\delta_{k,e} - \delta||^2\big] \leq 2E^2 L^2 \eta^2 \rho^2. \tag{B.1}$$

Where

$$\delta = \rho \frac{\nabla F(\boldsymbol{\theta}^t)}{||\nabla F(\boldsymbol{\theta}^t)||}, \qquad \delta_{k,e} = \rho \frac{\nabla F_k(\boldsymbol{\theta}^{t,e}, \xi_k)}{||\nabla F_k(\boldsymbol{\theta}^{t,e}, \xi_k)||}. \tag{B.2}$$

**Lemma 2 (Lemma B.2, Qu et al. (2022)).** Under the above assumptions, for any learning rate $\eta \leq \frac{1}{10EL}$, the updates exhibit drift arising from the deviation $\boldsymbol{\theta}^{t,e}(k) - \boldsymbol{\theta}^t$.

$$\frac{1}{K} \sum_k \mathbb{E}\big[||\boldsymbol{\theta}^{t,e}(k) - \boldsymbol{\theta}^t||^2\big] \leq 5E\eta^2 \Big( 2L^2\rho^2\sigma_l^2 + 6E(3\sigma_g^2 + 6L^2\rho^2) + 6E||\nabla F(\boldsymbol{\theta}^t)||^2 \Big) + 24K^3\eta^4 L^4 \rho^2. \tag{B.3}$$

**Lemma 3.** *The two model parameters obtained from two adjacent datasets differing by a single sample of client $k$ in communication round $t$,*

$$\sum_{e=0}^{E-1} ||y^{t,e}(i) - x^{t,e}(k)||_2^2 \leq 2E \max ||\Delta_k^t(y) - \Delta_k^t(x)||_2^2. \tag{B.4}$$

**Proof.** We recall that the local update performed by client $k$ is

$$\sum_{e=0}^{E-1} \boldsymbol{\theta}^{t,e}(k) = \sum_{e=0}^{E-1} \boldsymbol{\theta}^{t,e-1}(k) + \Delta_k^t,$$

(the initial value is assumed as $\boldsymbol{\theta}^{t-1} = \boldsymbol{\theta}^{t,0} = \boldsymbol{\theta}^t$). Then,

$$\sum_{k=0}^{E-1} ||y^{t,e}(i) - x^{t,e}(k)||_2^2 \leq 2 \sum_{e=0}^{E-1} ||y^{t,e-1}(k) - x^{t,e-1}(k)||_2^2 \\ + 2||\Delta_k^t(y) - \Delta_k^t(x)||_2^2. \tag{B.5}$$

Unrolling the recursion from $\tau = 0$ to $e$ yields

$$\sum_{e=0}^{E-1} ||y^{t,e}(k) - x^{t,k}(k)||_2^2 \overset{a)}{\leq} 2E \max ||\Delta_k^t(y) - \Delta_k^t(x)||_2^2. \tag{B.6}$$

Where a) uses the initial value $\boldsymbol{\theta}^t(k) = x^{t,0}(k) = y^{t,0}(k)$ and $0 < k \leq E$.

**Lemma 4.** *Assuming 1 and 3 hold, the aggregated update obtained by averaging the clipped local updates of the selected clients is*

$$\mathbb{E}||\frac{1}{m}\sum_{i\in W^t}\widetilde{\Delta}_k^t||^2 \leq 3E\eta^2(L^2\rho^2 + B^2). \tag{B.7}$$

**Proof.**

$$\mathbb{E}||\frac{1}{m}\sum_{k\in W^t}\widetilde{\Delta}_k^t||^2 \leq \mathbb{E}||\frac{1}{m}\sum_{i\in W^t}\sum_{k=0}^{E-1}\eta g_k^{t,e}(k)\cdot\alpha_k^t||^2$$

$$\leq \frac{\eta^2}{m}\sum_{k\in W^t}\sum_{k=0}^{E-1}\mathbb{E}||\nabla F_K(\boldsymbol{\theta}^{t,e}(k)+\delta;\xi_k) - \nabla F_k(\boldsymbol{\theta}^{t,E}(k);\xi_k)$$

$$+ \nabla F_k(\boldsymbol{\theta}^{t,e}(k);\xi_k) - \nabla F_k(\boldsymbol{\theta}^t(k)) + \nabla F_k(\boldsymbol{\theta}^t(k))||^2 \tag{B.8}$$

$$\overset{a)}{\leq} 3E\eta^2(L^2\rho^2 + B^2),$$

where (a) is derived using Assumptions 1 and 3, and

$$\alpha_k^t := \min\left(1, \frac{C}{\eta||\sum_{e=0}^{E-1}g_k^{t,e}(k)||_2}\right). \tag{B.9}$$

# B PROOF OF CONVERGENCE ANALYSIS

**Proof of Theorem 3.** For ease of reference, we define the following notations:

$$\Delta_k^t = \eta\sum_{e=0}^{E-1}g_k^e(k)\cdot\alpha_k^t, \quad \overline{\Delta}_k^t = \eta\sum_{e=0}^{E-1}g_k^e(k)\cdot\overline{\alpha}^t, \tag{6}$$

where

$$\alpha_k^t = \min\left(1, \frac{C}{\eta\sum_{e=0}^{E-1}||g_k^e(k)||}\right), \quad \pi^t = \frac{1}{K}\sum_{k=1}^K\alpha_k^t, \quad \widehat{\alpha}^t = \frac{1}{K}\sum_{k=1}^K|\alpha_k^t - \pi^t|. \tag{7}$$

The Lipschitz continuity of $\nabla F$:

$$\mathbb{E}[F(\boldsymbol{\theta}^{t+1})] \leq \mathbb{E}[F(\boldsymbol{\theta}^t)] + \mathbb{E}\langle\nabla F(\boldsymbol{\theta}^t), \boldsymbol{\theta}^{t+1} - \boldsymbol{\theta}^t\rangle + \frac{L}{2}||\boldsymbol{\theta}^{t+1} - \boldsymbol{\theta}^t||^2$$

$$= \mathbb{E}[F(\boldsymbol{\theta}^t)] + \langle\nabla F(\boldsymbol{\theta}^t), \frac{1}{m}\sum_{i\in W^t}\Delta_i^t + z_i^t\rangle + \frac{L}{2}\Big|\Big|\frac{1}{m}\sum_{i\in W^t}\Delta_i^t + z_i^t\Big|\Big|^2 \tag{8}$$

$$= \mathbb{E}[F(\boldsymbol{\theta}^t)] + \langle\nabla F(\boldsymbol{\theta}^t), \frac{1}{m}\sum_{i\in W^t}\Delta_i^t\rangle + \frac{L}{2}\Big|\Big|\frac{1}{m}\sum_{i\in W^t}\Delta_i^t\Big|\Big|^2 + \frac{L^2\sigma^2 C^2 pd}{2m^2}.$$

Here, $d$ is the dimension of $\boldsymbol{\theta}_k^t$, $p$ denotes the sparsity ratio, and the noise $z_k^t$ is assumed to have zero mean. Next, we analyze I and II in turn.

For I, we have

$$\langle\nabla F(\boldsymbol{\theta}^t), \frac{1}{m}\sum_{k\in W^t}\Delta_k^t\rangle = \langle\nabla F(\boldsymbol{\theta}^t), \frac{1}{K}\sum_{k=1}^K\Delta_k^t + \frac{1}{K}\sum_{k=1}^K(\Delta_k^t - \overline{\Delta}_k^t)\rangle + \langle\nabla F(\boldsymbol{\theta}^t), \frac{1}{K}\sum_{k=1}^K\overline{\Delta}_k^t\rangle. \tag{9}$$

Next, we provide bounds for the two terms in the preceding equality. In particular, for the first term we obtain

$$\mathbb{E}\langle\nabla F(\boldsymbol{\theta}^t), \frac{1}{E}\sum_{e=1}^E\Delta_e^t - \overline{\Delta}_e^t\rangle \leq \mathbb{E}\langle\nabla F(\boldsymbol{\theta}^t), \frac{1}{E}\sum_{e=1}^E\sum_{e=0}^{E-1}\eta\alpha_k^t\nabla F(\boldsymbol{\theta}^t, g_k^e(k))\rangle$$

$$\leq \frac{\eta E}{K}\sum_{k=1}^K\mathbb{E}\alpha_k^t\pi^t\langle\nabla F(\boldsymbol{\theta}^t), g_k^e(k)\rangle \tag{10}$$

$$\leq \frac{\eta^2 E}{2K}\sum_{k=1}^K\alpha_k^t\pi^t\Big(-\frac{1}{2}||\nabla F(\boldsymbol{\theta}^t)||^2 + ||\nabla F(\boldsymbol{\theta}^t, g_k^e(k)) - \nabla F(\boldsymbol{\theta}^t)||^2\Big)$$

$$\leq \eta^2 E(\widehat{\alpha}^t L^2\rho^2 - B^2),$$

where $\widehat{\alpha}^t = \frac{1}{K}\sum_{k=1}^{K}|\alpha_k^t - \pi^t|$. Here, (a) uses (a), (b) uses (b), and the result is established under Assumption 1.3.

For the second term, we have

$$\langle \nabla F(\boldsymbol{\theta}^t), \frac{1}{K}\sum_{k=1}^{K}\overline{\Delta}_k^t\rangle \le -\frac{\pi^t\eta E}{2}||\nabla F(\boldsymbol{\theta}^t)||^2 - \frac{\pi^t}{2E}||\frac{1}{K}\sum_{k=1}^{K}\Delta_k^t||^2$$
$$+ \frac{\pi^t}{2}||\nabla F(\boldsymbol{\theta}^t)||^2 - \frac{1}{\pi^t\eta EK}\sum_{k=1}^{K}||\overline{\Delta}_k^t||^2. \tag{11}$$

where $(a, b)$ employs the relation $(a, b) = ||a||^2 + ||b||^2 + \frac{1}{2}||a - b||^2$, under the condition $0 < \eta < 1$. Subsequently, we bound III as follows:

$$\text{III} = \frac{L}{2}||\nabla F(\boldsymbol{\theta}^t) + \frac{1}{K}\sum_{k=1}^{K}\frac{1}{E}\sum_{e=0}^{E-1}(\nabla F_k(\boldsymbol{\theta}_k^e; \xi_k) - \nabla F_k(\boldsymbol{\theta}^t))||^2$$
$$\le \frac{1}{K}\sum_{k=1}^{K}\frac{1}{E}\sum_{e=0}^{E-1}\mathbb{E}||\nabla F_k(\boldsymbol{\theta}_k^e + \delta_k) - \nabla F_k(\boldsymbol{\theta}^t)||^2$$
$$+ \eta(\nabla F_k(\boldsymbol{\theta}_k^e; \xi_k) - \nabla F_k(\boldsymbol{\theta}^t)) + (1 + \eta)\nabla F(\boldsymbol{\theta}^t)||^2 \tag{12}$$
$$\le 3\eta^2 E^2 L^2(\rho^2 + ||\nabla F(\boldsymbol{\theta}^t)||^2 + B^2)$$
$$+ 3\eta^2 E^2 L^2(2L^2\rho^2 + 6E(3\sigma_g^2 + 6L^2\rho^2) + 6E||\nabla F(\boldsymbol{\theta}^t)||^2)$$
$$+ 24E^2\eta^4 L^2\rho^2 + B^2,$$

where $0 < \eta < 1$, $(a, b)$, and the derivation relies on Assumptions 1 and 3 as well as Lemma 3, respectively. For the second term (II), we invoke Lemma 8. By combining Eqs. 12–16, we arrive at

$$\mathbb{E}F(\boldsymbol{\theta}^{t+1}) \le \mathbb{E}F(\boldsymbol{\theta}^t) + \eta\pi^t EL^2(\rho^2 - B^2) - \frac{\pi^t\eta E}{2}||\nabla F(\boldsymbol{\theta}^t)||^2$$
$$- \frac{\eta\pi^t}{2E}||\frac{1}{K}\sum_{k=1}^{K}\Delta_k^t||^2$$
$$+ \frac{3\pi^t\eta^2 E}{2}\left[2L^2\rho^2\sigma_k^2 + 6E(3\sigma_g^2 + 6L^2\rho^2) + 6E||\nabla F(\boldsymbol{\theta}^t)||^2\right] \tag{13}$$
$$+ \frac{3\pi^t\eta^2 E}{2}(3\eta^2 EL^2\rho^2 + B^2) + \frac{3\eta^2 EL(L^2\rho^2 + B^2)}{2}$$
$$+ \frac{L\sigma^2 C^2 pd}{2m^2}.$$

When $\eta \le \frac{1}{3\sqrt{EL}}$, the inequality is

$$\mathbb{E}F(\boldsymbol{\theta}^{t+1}) \le F(\boldsymbol{\theta}^t) - \frac{\pi^t\eta E}{2}||\nabla F(\boldsymbol{\theta}^t)||^2 + \frac{\eta^2 EL^2\rho^2}{2}$$
$$+ \frac{3\pi^t\eta^2 EL^2\rho^2}{2} + \frac{15\pi^t E\eta^2 L^2}{2} \tag{30}$$
$$+ \frac{3\pi^t\eta^2 EL(L^2\rho^2 + B^2)}{2} + \frac{L\sigma^2 C^2 pd}{2m^2}.$$

Sum over $t$ from 1 to $T$, we have

$$\frac{1}{T}\sum_{t=1}^{T}\mathbb{E}||\nabla F(\boldsymbol{\theta}^t)||^2 \le \frac{2L(F(\boldsymbol{\theta}^1) - F^*)}{\sqrt{ET}} + \frac{1}{T}\sum_{t=1}^{T}\left(2\pi^t EL^2\rho^2 - 2a\widehat{\alpha}^t B^2 + 30\eta^2 L^2 T\right)$$
$$+ \frac{1}{T}\sum_{t=1}^{T}\left(2L^2\rho^2\sigma_k^2 + 6E(3\sigma_g^2 + 6L^2\rho^2)\right) \tag{14}$$
$$+ 72\eta^4 E^3 L^2\rho^2 + 3\eta L(L^2\rho^2 + B^2) + \frac{L\sigma^2 C^2 pd}{m^2 E}.$$

Assume the local adaptive learning rate satisfies $\eta = O(1/L\sqrt{KT})$, both $\sum_{t=1}^{T}\widehat{\alpha}^t$ and $\sum_{t=1}^{T}\bar{\alpha}^t$ are two important parameters for measuring the impact of clipping. Meanwhile, both $\frac{1}{T}\sum_{t=1}^{T}\widehat{\alpha}^t$ and $\frac{1}{T}\sum_{t=1}^{T}\bar{\alpha}^t$ are also bounded by 1. Then, our result is

$$\frac{1}{T}\sum_{t=1}^{T}\mathbb{E}\big[\pi^t||\nabla F(\boldsymbol{\theta}^t)||^2\big] \leq O\left(\frac{2L(F(\boldsymbol{\theta}^1) - F^*)}{\sqrt{ET}} + \left(\frac{1}{T}\sum_{t=1}^{T}\frac{\sigma_g^2}{T^2} + \frac{L^2\rho^2}{T^2}\right) + \frac{L^2\sigma^2 C^2 pd}{m^2\sqrt{E}}\right). \quad (15)$$

If the perturbation amplitude $\rho$ is chosen proportional to the learning rate, such as $\rho = O(1/\sqrt{T})$, it follows that

$$\frac{1}{T}\sum_{t=1}^{T}\mathbb{E}||\nabla F(\boldsymbol{\theta}^t)||^2 \leq O\left(\frac{2L(F(\boldsymbol{\theta}^1) - F^*)}{\sqrt{ET}} + \frac{L^2\rho^2}{ET^2} + \left(\frac{1}{T}\sum_{t=1}^{T}\frac{\sigma_g^2}{T^2} + \frac{L^2\rho^2}{T^2}\right) + \frac{L\sigma^2 C^2 pd}{m^2\sqrt{E}}\right). \quad (16)$$

## C  DIFFERENTIAL PRIVACY GUARANTEES IN FEDERATED LEARNING

We utilize Rényi Differential Privacy (RDP) as our primary privacy measure, which provides strong composition properties suitable for iterative federated learning procedures.

### C.1  RÉNYI DIFFERENTIAL PRIVACY (RDP)

**Definition C.1** (Rényi Differential Privacy Abadi et al. (2016)). A randomized mechanism $\mathcal{M}$ satisfies $(\alpha, \rho)$-RDP if for any two neighboring datasets $D, D'$ differing by a single client's data, the following holds:

$$D_\alpha(\mathcal{M}(D)||\mathcal{M}(D')) = \frac{1}{\alpha - 1}\log\mathbb{E}\left[\left(\frac{\mathcal{M}(D)}{\mathcal{M}(D')}\right)^\alpha\right] \leq \rho. \quad (17)$$

### C.2  SENSITIVITY ANALYSIS

The sensitivity of local updates is critical to determining the magnitude of added noise. Given local updates $g_k(\boldsymbol{\theta}_t)$ at round $t \in T$, sensitivity is defined as:

$$S_{g_k(\boldsymbol{\theta}_t)}^2 = \max_{D \simeq D'}||g_k(\boldsymbol{\theta}_t)(D) - g_k(\boldsymbol{\theta}_t)(D')||_2^2. \quad (18)$$

Utilizing SAM optimization as in Shi et al. (2023), we have the sensitivity bound for SAM updates:

$$\mathbb{E}[S_{g_k(\boldsymbol{\theta}_t)}^2] \leq O\left(\frac{1}{T^2}\right), \quad (19)$$

### C.3  GAUSSIAN MECHANISM

We employ the Gaussian mechanism to perturb clipped local updates. Given a clipping threshold $C$, noise is added to the local updates as follows:

$$\tilde{g}_k(\boldsymbol{\theta}_t) \leftarrow g_k(\boldsymbol{\theta}_t) \cdot \min\left(1, \frac{C}{||g_k(\boldsymbol{\theta}_t)||_2}\right) + \mathcal{N}\left(0, \frac{\sigma^2 C^2}{m}I_d\right), \quad (20)$$

where $m$ is the number of participating clients per round, and $\sigma^2$ is the variance of the Gaussian noise.

### C.4  PRIVACY BUDGET (CUMULATIVE PRIVACY LOSS)

The cumulative privacy budget $\epsilon$ after $T$ rounds is derived from the Gaussian mechanism via RDP composition as follows:

$$\epsilon = \epsilon' + \frac{(\alpha - 1)\log(1 - \frac{1}{\alpha}) - \log(\alpha) - \log(\delta)}{\alpha - 1}, \quad (21)$$

where

$$\epsilon' = \frac{T}{\alpha - 1}\log\mathbb{E}_{z\sim\mu_0}\left[\left(1 - q + q\frac{\mu_1(z)}{\mu_0(z)}\right)^\alpha\right]. \quad (22)$$

Here, $q = \frac{m}{M}$ represents the client sampling ratio, and $\mu_0, \mu_1$ denote the Gaussian probability density functions for distributions $\mathcal{N}(0, \sigma)$ and the mixture $q\mathcal{N}(1, \sigma) + (1 - q)\mathcal{N}(0, \sigma)$, respectively. Parameter $\alpha$ is selected to optimize the privacy-utility trade-off.

# D    PRELIMINARIES AND PROBLEM SETUP

The purpose of this section is to provide additional *Preliminaries and Problem Setup* details that are dropped due to the limited space of the main paper. It includes the plots of training instability and the Experimental Setup for the CIFAR10, CIFAR100, and UTK datasets.

## D.1    TRAINING INSTABILITY STUDY: RESULTS OVERVIEW

We measure the training instability (relative progresss $RP$) and its impacts on generalization (utility/accuracy) in FL across CIFAR10 (Figure 3), UTK (Figure 4,) and CIFAR100 (Figure 5) datasets. As depicted in most of the graphs on the left, the $RP$, which measures the stability of training, shows significant variance across training rounds. This variance is persistent and positive, indicating that the learning process is not stable. This instability is further compounded as the differential privacy level increases. The increasing variance in $RP$ with higher levels of DP suggests that the noise added for privacy protection is disrupting the learning process, making it harder for the model to converge consistently. This behavior demonstrates the challenge of balancing model privacy with learning efficacy in FL environments.

The middle graphs show $RP$ variance against different levels of differential privacy and confirm that as privacy constraints tighten ($\sigma^2$ increases), the overall variability in model performance also increases. The trend line indicates a clear positive correlation between $RP$ variance and the privacy level, highlighting a direct impact of enhanced privacy measures on learning stability. Higher differential privacy levels introduce more noise into the training process, which can lead to larger updates that are less about the true gradient direction and more about compensating for the noise. This can cause the training process to become unstable, as shown by the rising $RP$ variance.

The right most graphs illustrates that with increasing DP, not only do gradient norms increase, but also accuracy decreases significantly. This suggests that the model is struggling to generalize effectively under higher training instability. Larger gradient norms indicate more substantial updates during training, which can overshoot optimal points due to the high noise levels introduced by DP. This is likely contributing to the observed decrease in model accuracy as DP levels increase, illustrating the difficulty in navigating the trade-off between privacy and generalization.

These detailed analyses and observations demonstrate the complex interplay between privacy, stability, and generalization in FL. By fine-tuning the learning rates and understanding the impact of differential privacy on learning dynamics, it is possible to improve both the stability and generalization of models trained under privacy constraints.

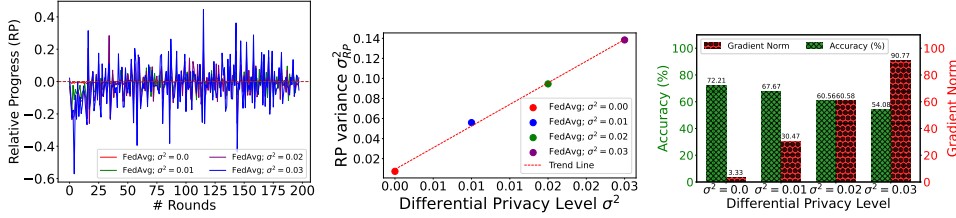

Figure 3: Correlation of DP noise with training instability in a 10-client CIFAR10 setup ($\alpha = 0.3$ non-iid, $\eta_0 = 0.1$). Increased DP noise elevates instability, as shown by RP value variance, causing larger gradient norms and lower accuracy.

## D.2    EXPERIMENTAL SETUP

### D.2.1    DATASETS AND MODEL ARCHITECTURES

Table 5: Datasets and Clients

| Dataset | Task | Total Clients | Total Samples | Training Samples | Test Samples |
|---|---|---|---|---|---|
| CIFAR10 | Image classification | 10,20 | 60,000 | 50,000 | 10,000 |
| CIFAR100 | Image classification | 10,20 | 60,000 | 50,000 | 10,000 |
| UTK | Image classification | 10,20 | 23,708 | 19,208 | 4,500 |

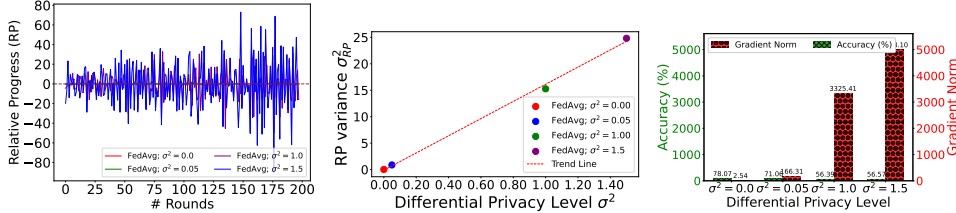

Figure 4: Correlation of DP noise with training instability in a 10-client UTK setup ($\alpha = 0.3$ non-iid, $\eta_0 = 0.1$). Increased DP noise elevates instability, as shown by RP value variance, causing larger gradient norms and lower accuracy.

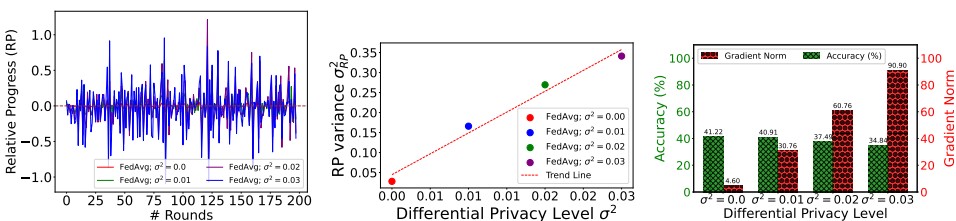

Figure 5: Correlation of DP noise with training instability in a 10-client CIFAR100 setup ($\alpha = 0.3$ non-iid, $\eta_0 = 0.1$). Increased DP noise elevates instability, as shown by RP value variance, causing larger gradient norms and lower accuracy.

**Dataset:** We analyze two widely utilized image classification datasets for federated learning: CIFAR10 Krizhevsky et al. (2009) and CIFAR100 Sharma et al. (2018), along with the UTK Savchenko (2021) image classification dataset. The benchmarks for these datasets in a federated learning context are adopted from established benchmarks based on CIFAR-10/100, as proposed by Foret et al. (2020). Each dataset is allocated among $K \in \{10, 20\}$ clients, employing a Dirichlet distribution-based approach for data distribution as done in Zeng et al. (2023). The resultant data partitions are shown in Figure 6, Figure 7, Figure 8, and Figure 9. Here, each client's prior distribution follows a multinomial distribution derived from a symmetric Dirichlet distribution with parameter $\alpha$. As $\alpha$ approaches infinity, the data distribution among clients approximates an IID scenario. Conversely, a reduction in $\alpha$, moving towards zero, shifts the distribution towards a non-IID scenario. We explore different scenarios with $\alpha \in \{0.05, 0.3\}$ across the CIFAR10, CIFAR100, and UTK datasets.

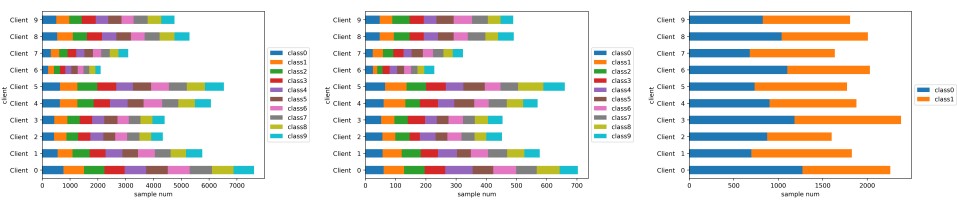

Figure 6: Noniid partition used in Yurochkin et al. (2019) and Wang et al. (2020a). The number of CIFAR10, CIFAR1OO, and UTK data points and class proportions are unbalanced. Samples will be partitioned into 10 clients by sampling $\alpha = 0.3$.

**Model architecture:** By following the backbone architecture of the unstable convergence of gradient descent work Ahn et al. (2022); Cohen et al. (2021). Specifically, we use GD to train a VGG (with batch normalization) neural network Ding et al. (2021). For a fair comparison, we use the same backbone architecture for all different types of methods for all evaluations. Also, the same architecture is identically used for the two CIFAR-10/100 benchmarks. Noteworthy, we added ResNet backbone for UTK dataset because of poor performance relative to VGG on this dataset.

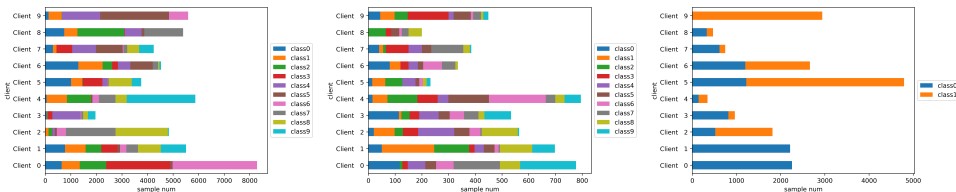

Figure 7: Noniid partition used in Yurochkin et al. (2019) and Wang et al. (2020a). Number of CIFAR10, CIFAR1OO, and UTK data points and class proportions are unbalanced. Samples will be partitioned into 10 clients by sampling $\alpha = 0.05$.

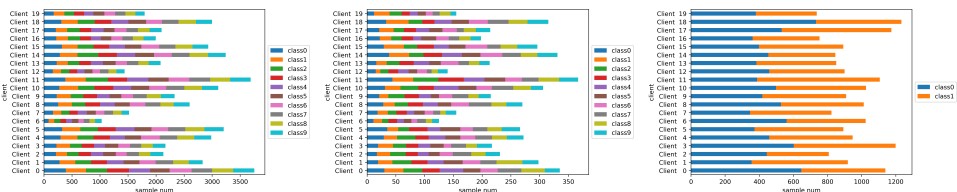

Figure 8: Noniid partition used in Yurochkin et al. (2019) and Wang et al. (2020a). The number of CIFAR10, CIFAR1OO, and UTK data points and class proportions are unbalanced. Samples will be partitioned into 20 clients by sampling $\alpha = 0.3$.

### D.2.2 DATA PRE-PROCESSING (CIFAR-10 AND CIFAR-100).

All training and test input images of size $32 \times 32$ pixels are first padded by 4 pixels on each side, then randomly cropped back to $32 \times 32$ pixels. This technique helps the model become invariant to small translations of the input image. Each image is flipped horizontally with a probability of $0.5$. This step increases the diversity of the training data and helps prevent overfitting by simulating different viewing angles. After converting the image to a tensor, pixel values are normalized using the dataset-specific mean $(0.4914, 0.4822, 0.4465)$ and standard deviation $(0.2023, 0.1994, 0.2010)$. This normalization facilitates faster convergence by scaling the input features to have zero mean and unit variance.

### D.2.3 DATA PRE-PROCESSING (UTK).

All training and test input images are resized to $32 \times 32$ pixels, standardizing the input size across all images and making it suitable for processing by the model designed for CIFAR datasets. Pixel values are normalized using the mean $(0.49)$ and standard deviation $(0.23)$. This dataset appears to have grayscale images (indicated by a single channel mean and standard deviation), and normalization adjusts the pixel intensity distribution similarly to CIFAR datasets. Images undergo the same resizing to $32 \times 32$ pixels and are normalized using the same values as the training images. Consistent image size and normalization between the training and testing phases help in evaluating the model's performance accurately.

## E ADDITIONAL EXPERIMENTAL RESULTS

Here, we provide the additional experimental results that are dropped due to the limited space of the main paper. It includes the the plots for *generalization analysis*, *rate of convergence analysis*, and *training stability analysis* using for the CIFAR10, CIFAR100, and UTK datasets.

### E.1 GENERALIZATION ANALYSIS FL

We conduct a thorough analysis of ADAPT-FED's generalization performance against various baseline FL algorithms. Our primary goal is to assess the efficacy of ADAPT-FED in generalizing under diverse privacy settings and heterogeneous data distributions. Generalization analyses are performed on three widely recognized datasets: CIFAR10, CIFAR100, and UTK, comparing ADAPT-FED with several SOTA FL algorithms, including FedAvg, FedProx, FedAdagrad, FedYogi, FedSAM, and FedASAM.

Table 6 shows the results with 10 clients and learning rate $\eta_o = 0.1$. ADAPT-FED consistently outperforms all baselines under both low and high heterogeneity settings across all three datasets and all privacy noise levels. Notably, its performance remains stable even as $\varepsilon$ increases, highlighting its robustness under privacy constraints.

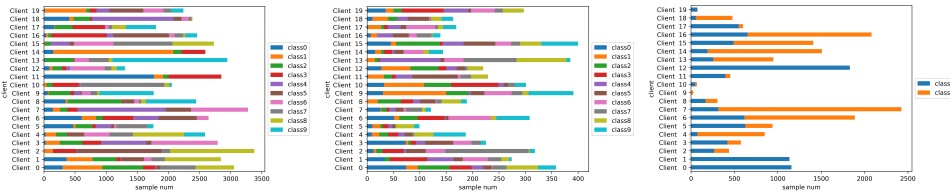

Figure 9: Noniid partition used in Yurochkin et al. (2019) and Wang et al. (2020a). The number of CIFAR10, CIFAR1OO, and UTK data points and class proportions are unbalanced. Samples will be partitioned into 20 clients by sampling $\alpha = 0.05$.

Table 7 reports results for 20 clients at $\eta_o = 0.04$. Here too, ADAPT-FED achieves the highest test accuracy across all settings. Compared to baselines, its margins of improvement are particularly significant under high non-IID scenarios and when privacy noise is introduced. Table 8 further confirms the generalization advantage of ADAPT-FED under similar settings ($\eta_k = 0.04$, $C = 100$). The performance trends remain consistent across the three datasets. Finally, Table 9 explores two extreme participation scenarios: full clipping constants $C = 100$ and $C = 1$. ADAPT-FED demonstrates remarkable stability and maintains generalization advantage in both cases. Unlike FedAvg and FedProx, which degrade significantly under $C = 1$, ADAPT-FED remains effective even under extreme clipping and high noise, suggesting its suitability for deployment in practical, resource-constrained FL settings.

Across all experiments, ADAPT-FED achieves the highest or near-highest accuracy in nearly every configuration, supporting its efficacy for privacy-preserving, prompt-based federated learning in heterogeneous environments.

Table 6: Generalization performance of ADAPT-FED versus baseline algorithms based on 10 clients across three datasets: (a) CIFAR10, (b) CIFAR100, and (c) UTK, respectively, $\eta_o = 0.1$. ADAPT-FED outperforms the baseline algorithms in terms of generalization performance across datasets.

| Algorithm | CIFAR-10 | | | | | | | | CIFAR-100 | | | | | | | | UTK | | | | | | | |
|---|---|---|---|---|---|---|---|---|---|---|---|---|---|---|---|---|---|---|---|---|---|---|---|---|
| | Dir. ($\alpha = 0.05$, non-IID) | | | | Dir. ($\alpha = 0.3$) | | | | Dir. ($\alpha = 0.05$, non-IID) | | | | Dir. ($\alpha = 0.3$) | | | | Dir. ($\alpha = 0.05$, non-IID) | | | | Dir. ($\alpha = 0.3$) | | | |
| | 0.0 | 0.01 | 0.02 | 0.03 | 0.0 | 0.01 | 0.02 | 0.03 | 0.0 | 0.01 | 0.02 | 0.03 | 0.0 | 0.01 | 0.02 | 0.03 | 0.0 | 0.01 | 0.02 | 0.03 | 0.0 | 0.01 | 0.02 | 0.03 |
| FedAvg | 58.76 | 53.72 | 47.31 | 41.91 | 74.55 | 70.18 | 64.76 | 59.68 | 37.73 | 37.03 | 33.94 | 30.89 | 41.21 | 40.91 | 37.48 | 34.84 | 64.11 | 64.11 | 55.18 | 54.98 | 78.89 | 78.89 | 64.87 | 64.83 |
| FedSAM | 57.93 | 58.08 | 58.67 | 58.74 | 74.80 | 74.18 | 74.61 | 74.90 | 37.43 | 38.49 | 38.57 | 38.17 | 42.18 | 42.64 | 42.81 | 43.89 | 73.28 | 73.28 | 73.58 | 73.35 | 79.06 | 79.06 | 79.10 | 78.83 |
| FedASAM | 59.09 | 58.78 | 58.74 | 58.74 | 75.40 | 74.91 | 74.51 | 74.51 | 38.56 | 37.83 | 37.99 | 37.99 | 43.39 | 43.12 | 43.28 | 43.28 | 73.31 | 73.31 | 74.07 | 74.07 | 78.63 | 78.63 | 79.48 | 79.48 |
| FedProx | 59.86 | 55.27 | 49.14 | 42.82 | 73.54 | 69.48 | 64.36 | 59.91 | 37.90 | 36.37 | 34.30 | 30.97 | 42.36 | 40.66 | 36.87 | 34.84 | 65.42 | 65.42 | 55.05 | 54.68 | 71.40 | 71.40 | 57.03 | 56.65 |
| FedAdagrad | 58.76 | 53.72 | 47.31 | 41.91 | 74.55 | 70.18 | 64.76 | 59.68 | 37.73 | 37.03 | 33.94 | 30.89 | 41.21 | 40.91 | 37.48 | 34.84 | 64.11 | 64.11 | 55.18 | 54.98 | 71.06 | 71.06 | 56.38 | 56.56 |
| FedAdam | 58.76 | 53.72 | 47.31 | 41.91 | 74.55 | 70.18 | 64.76 | 59.68 | 37.73 | 37.03 | 33.94 | 30.89 | 41.21 | 40.91 | 37.48 | 34.84 | 64.11 | 64.11 | 55.18 | 54.98 | 78.89 | 78.89 | 64.87 | 64.83 |
| FedYogi | 58.76 | 53.72 | 47.31 | 41.91 | 74.55 | 70.18 | 64.76 | 59.68 | 37.73 | 37.03 | 33.94 | 30.89 | 41.21 | 40.91 | 37.48 | 34.84 | 64.11 | 64.11 | 55.18 | 54.98 | 78.89 | 78.89 | 64.87 | 64.83 |
| ADAPT-FED (ours) | 63.02 | 65.18 | 65.39 | 65.83 | 80.46 | 81.33 | 81.24 | 81.75 | 51.44 | 53.59 | 54.26 | 54.34 | 58.18 | 61.25 | 61.38 | 60.39 | 75.05 | 75.05 | 75.05 | 74.31 | 86.85 | 86.85 | 86.49 | 86.86 |

Table 7: Generalization performance of ADAPT-FED versus baseline algorithms based on 20 clients across three datasets: (a) CIFAR10, (b) CIFAR100, and (c) UTK, respectively, $\eta_o = 0.04$.

| Algorithm | CIFAR-10 | | | | | | | | CIFAR-100 | | | | | | | | UTK | | | | | | | |
|---|---|---|---|---|---|---|---|---|---|---|---|---|---|---|---|---|---|---|---|---|---|---|---|---|
| | Dir. ($\alpha = 0.05$, non-IID) | | | | Dir. ($\alpha = 0.3$) | | | | Dir. ($\alpha = 0.05$, non-IID) | | | | Dir. ($\alpha = 0.3$) | | | | Dir. ($\alpha = 0.05$, non-IID) | | | | Dir. ($\alpha = 0.3$) | | | |
| | 0.0 | 0.01 | 0.02 | 0.03 | 0.0 | 0.01 | 0.02 | 0.03 | 0.0 | 0.01 | 0.02 | 0.03 | 0.0 | 0.01 | 0.02 | 0.03 | 0.0 | 0.01 | 0.02 | 0.03 | 0.0 | 0.01 | 0.02 | 0.03 |
| FedAvg | 50.63 | 48.59 | 44.62 | 40.86 | 67.48 | 66.28 | 62.84 | 59.72 | 33.09 | 32.61 | 30.82 | 29.26 | 50.10 | 49.61 | 47.38 | 45.07 | 70.58 | 70.58 | 53.86 | 53.08 | 80.37 | 80.37 | 59.29 | 58.26 |
| FedSAM | 50.47 | 50.73 | 51.03 | 51.11 | 67.59 | 66.66 | 67.25 | 67.42 | 32.97 | 33.32 | 34.09 | 33.24 | 50.75 | 50.71 | 50.61 | 51.03 | 75.59 | 75.59 | 76.39 | 76.54 | 83.75 | 83.75 | 83.75 | 83.59 |
| FedASAM | 51.26 | 51.48 | 51.23 | 51.23 | 67.06 | 67.98 | 67.63 | 67.63 | 34.10 | 33.94 | 33.85 | 33.85 | 51.04 | 51.04 | 50.74 | 50.74 | 76.19 | 76.19 | 77.03 | 77.03 | 83.69 | 83.69 | 83.81 | 83.81 |
| FedProx | 50.82 | 48.78 | 44.77 | 41.15 | 66.97 | 65.27 | 62.45 | 59.17 | 32.98 | 32.70 | 31.00 | 29.42 | 49.67 | 49.47 | 47.51 | 44.68 | 70.49 | 70.49 | 53.59 | 53.12 | 79.75 | 79.75 | 58.94 | 57.02 |
| FedAdagrad | 50.89 | 48.66 | 44.68 | 40.96 | 67.48 | 66.28 | 62.84 | 59.72 | 32.98 | 32.70 | 31.00 | 29.42 | 50.10 | 49.61 | 47.38 | 45.07 | 70.58 | 70.58 | 53.86 | 53.08 | 80.35 | 80.84 | 59.94 | 58.62 |
| FedAdam | 50.63 | 48.59 | 44.62 | 40.86 | 67.48 | 66.28 | 62.84 | 59.72 | 32.98 | 32.70 | 31.00 | 29.42 | 50.10 | 49.61 | 47.38 | 45.07 | 70.58 | 70.58 | 53.86 | 53.08 | 80.45 | 80.91 | 59.61 | 58.10 |
| FedYogi | 50.63 | 48.59 | 44.62 | 40.86 | 67.48 | 66.28 | 62.84 | 59.72 | 32.98 | 32.70 | 31.00 | 29.42 | 50.10 | 49.61 | 47.38 | 45.07 | 70.58 | 70.58 | 53.86 | 53.08 | 80.37 | 80.37 | 59.29 | 58.26 |
| ADAPT-FED (ours) | 56.49 | 59.66 | 59.78 | 60.01 | 75.86 | 77.48 | 77.65 | 77.99 | 40.66 | 44.63 | 43.52 | 44.51 | 54.69 | 57.40 | 49.83 | 54.27 | 81.31 | 81.31 | 80.15 | 81.53 | 85.56 | 85.56 | 84.15 | 84.67 |

## E.2 RATE OF CONVERGENCE ANALYSIS

We conduct a thorough analysis of ADAPT-FED's convergence performance against various baseline FL algorithms. Our primary goal is to assess the efficacy of ADAPT-FED in achieving faster and more stable convergence rates, particularly under diverse privacy settings and heterogeneous data distributions. Convergence analyses are performed on three widely recognized datasets: CIFAR-10, CIFAR-100, and UTK, comparing ADAPT-FED with SOTA FL algorithms, including FedAvg, FedSAM, and FedASAM.

As illustrated in Figure 2, ADAPT-FED demonstrates robust convergence in settings with data heterogeneity ($\alpha = 0.3$). This performance is indicative of the adaptive learning rate mechanism within ADAPT-FED, which fine-tunes the updates based on the observed instability and heterogeneity levels, thereby enhancing the convergence rate.

ADAPT-FED utilizes an innovative adaptive learning rate strategy that dynamically adjusts based on the model's performance from one iteration to the next. This approach addresses not only the variability introduced by differential privacy but also the challenges posed by non-IID data across clients. Unlike traditional methods that

Table 8: Generalization performance of ADAPT-FED versus baseline algorithms based on 20 clients across three datasets: (a) CIFAR10, (b) CIFAR100, and (c) UTK, respectively, $\eta_k \in \{0.04\}, C = 100$.

| Algorithm | CIFAR-10 | | | | | | | | CIFAR-100 | | | | | | | | UTK | | | | | | | |
|---|---|---|---|---|---|---|---|---|---|---|---|---|---|---|---|---|---|---|---|---|---|---|---|---|
| | Dir. ($\alpha = 0.05$, non-IID) | | | | Dir. ($\alpha = 0.3$) | | | | Dir. ($\alpha = 0.05$, non-IID) | | | | Dir. ($\alpha = 0.3$) | | | | Dir. ($\alpha = 0.05$, non-IID) | | | | Dir. ($\alpha = 0.3$) | | | |
| | 0.0 | 0.01 | 0.02 | 0.03 | 0.0 | 0.01 | 0.02 | 0.03 | 0.0 | 0.01 | 0.02 | 0.03 | 0.0 | 0.01 | 0.02 | 0.03 | 0.0 | 0.01 | 0.02 | 0.03 | 0.0 | 0.01 | 0.02 | 0.03 |
| FedAvg | 50.63 | 48.59 | 44.62 | 40.86 | 67.48 | 66.28 | 62.84 | 59.72 | 33.09 | 32.61 | 30.82 | 29.26 | 50.10 | 49.61 | 47.38 | 45.07 | 70.58 | 70.58 | 53.86 | 53.08 | 80.37 | 80.37 | 59.29 | 58.26 |
| FedSAM | 50.47 | 50.73 | 51.03 | 51.11 | 67.59 | 66.66 | 67.25 | 67.42 | 32.97 | 33.32 | 34.09 | 33.24 | 50.75 | 50.71 | 50.61 | 51.03 | 75.59 | 75.59 | 76.39 | 76.54 | 83.75 | 83.75 | 83.75 | 83.59 |
| FedASAM | 51.26 | 51.48 | 51.23 | 51.23 | 67.06 | 67.98 | 67.63 | 67.63 | 34.10 | 33.94 | 33.85 | 33.85 | 51.04 | 51.04 | 50.74 | 50.74 | 76.19 | 76.19 | 77.03 | 77.03 | 83.69 | 83.69 | 83.81 | 83.81 |
| FedProx | 50.82 | 48.78 | 44.77 | 41.15 | 66.97 | 65.27 | 62.45 | 59.17 | 32.98 | 32.70 | 31.00 | 29.42 | 49.67 | 49.47 | 47.51 | 44.68 | 70.49 | 70.49 | 53.59 | 53.12 | 79.75 | 79.75 | 58.94 | 57.02 |
| ADAPT-FED (ours) | 56.49 | 59.66 | 59.78 | 60.01 | 75.86 | 77.48 | 77.65 | 77.99 | 40.66 | 44.63 | 43.52 | 44.51 | 54.69 | 57.40 | 49.83 | 54.27 | 81.31 | 81.31 | 80.15 | 81.53 | 85.56 | 85.56 | 84.15 | 84.67 |

Table 9: Generalization performance of ADAPT-FED versus baseline algorithms based on 20 clients across three datasets: (a) CIFAR10, (b) CIFAR100, and (c) UTK, respectively, $\eta_k \in \{0.04\}, C = 100$.

| Algorithm | CIFAR-10 | | | | | | | | CIFAR-100 | | | | | | | | UTK | | | | | | | |
|---|---|---|---|---|---|---|---|---|---|---|---|---|---|---|---|---|---|---|---|---|---|---|---|---|
| | $C = 1$ | | | | $C = 100$ | | | | $C = 1$ | | | | $C = 100$ | | | | $C = 1$ | | | | $C = 100$ | | | |
| | 0.0 | 0.01 | 0.02 | 0.03 | 0.0 | 0.01 | 0.02 | 0.03 | 0.0 | 0.01 | 0.02 | 0.03 | 0.0 | 0.01 | 0.02 | 0.03 | 0.0 | 0.01 | 0.02 | 0.03 | 0.0 | 0.01 | 0.02 | 0.03 |
| FedAvg | 67.48 | 62.43 | 54.67 | 48.36 | 67.48 | 66.28 | 62.84 | 59.72 | 50.10 | fail | fail | fail | 50.10 | 49.61 | 47.38 | 45.07 | 80.37 | 79.05 | 51.48 | 50.45 | 80.37 | 80.37 | 59.29 | 58.26 |
| FedSAM | 67.59 | 68.08 | 67.80 | 68.25 | 67.59 | 66.66 | 67.25 | 67.42 | 50.75 | 50.82 | 50.48 | 50.72 | 50.75 | 50.71 | 50.61 | 51.03 | 83.75 | 83.38 | 83.79 | 83.75 | 83.75 | 83.75 | 83.75 | 83.59 |
| FedASAM | 67.06 | 68.15 | 68.29 | 68.29 | 67.06 | 67.98 | 67.63 | 67.63 | 51.04 | 50.60 | 51.01 | 51.01 | 51.04 | 51.04 | 50.74 | 50.74 | 83.69 | 83.98 | 83.56 | 83.56 | 83.69 | 83.69 | 83.81 | 83.81 |
| FedProx | 66.97 | 62.07 | 52.56 | 43.63 | 66.97 | 65.27 | 62.45 | 59.17 | 49.67 | fail | fail | fail | 49.67 | 49.47 | 47.51 | 44.68 | 79.75 | 71.31 | 48.74 | 48.79 | 79.75 | 79.75 | 58.94 | 57.02 |
| ADAPT-FED (ours) | 75.86 | 78.03 | 77.75 | 77.79 | 75.86 | 77.48 | 77.65 | 77.99 | 54.69 | 57.81 | 47.94 | 54.61 | 54.69 | 57.40 | 49.83 | 54.27 | 85.56 | 84.01 | 84.90 | 84.45 | 85.56 | 85.56 | 84.15 | 84.67 |

apply uniform updates, ADAPT-FED tailors the learning rates to mitigate the impact of high gradient variances and ensures consistent learning progress.

### E.3 TRAINING STABILITY ANALYSIS

We evaluate the training stability of ADAPT-FED in comparison to various baseline FL algorithms. These experiments are conducted across the CIFAR10, CIFAR100, and UTK datasets, with emphasis on differential privacy settings and data heterogeneity.

Figure 10, Figure 12, and Figure 11, illustrate the relative progress (RP) across 200 training round under varying conditions. These figures capture the effectiveness of ADAPT-FED's adaptive learning rate mechanism in enhancing training stability compared to traditional FL approaches. This strategy significantly reduces the oscillations in $RP$, particularly evident in scenarios with high differential privacy levels and heterogeneous data distributions. ADAPT-FED maintains a lower variance in RP compared to baselines like FedAvg and FedProx, indicating more consistent progress and reduced training disruptions despite the introduction of noise through differential privacy. Figure 10 Figure 12, and Figure 11 highlight ADAPT-FED's ability to sustain lower variability in $RP$ even under severe data heterogeneity, reflecting its capacity to adapt to heterogeneous data distributions effectively. ADAPT-FED employs an adaptive learning rate that dynamically adjusts based on the observed gradient norms.

While baseline algorithms exhibit increased $RP$ fluctuations, indicating struggles with gradient noise and data heterogeneity, ADAPT-FED demonstrates a markedly smoother convergence curve. This distinction demonstrates the limitations of SOTA methods that do not account dynamically for changing gradient scales, often leading to inefficient learning rates that either overstep or underutilize the learning potential of the model.

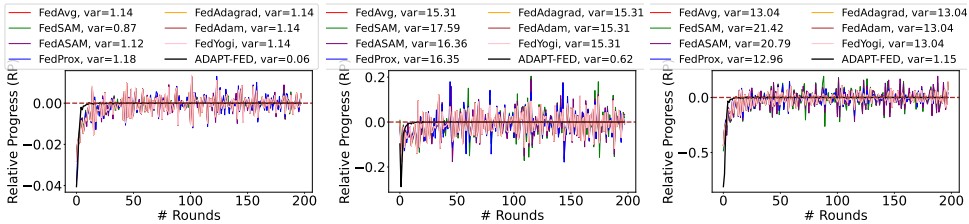

Figure 10: Stability of the training loss of ADAPT-FED and baseline algorithms on 10 clients (CIFAR10 noniid-ness $\alpha = 0.3$, $\eta_0 = 0.1$) across three DP levels: (a) $\sigma^2 = 0.0$, (b) $\sigma^2 = 0.01$, and (c) $\sigma^2 = 0.02$, respectively. ADAPT-FED exhibits more stable convergence compared to baselines.

## F ALTERNATIVE RP-BASED $\beta$ SCHEDULING MECHANISMS

To address the heuristic nature of the exponential $RP$ transformation, we explore several theoretically motivated alternatives to the $\beta$ function in ADAPT-FED. These variants aim to improve the robustness, interpretability, and adaptability of local learning rate schedules under varying sharpness and instability conditions.

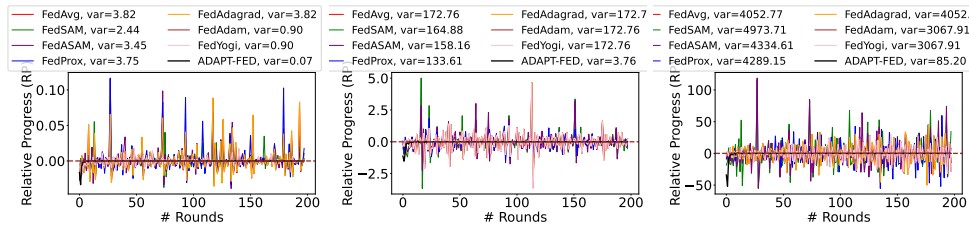

Figure 11: Stability of the training loss of ADAPT-FED and baseline algorithms on 10 clients (UTK noniid-ness $\alpha = 0.3$, $\eta_0 = 0.1$) across three DP levels: (a) $\sigma^2 = 0.0$, (b) $\sigma^2 = 0.01$, and (c) $\sigma^2 = 0.02$, respectively. ADAPT-FED exhibits more stable convergence compared to baselines.

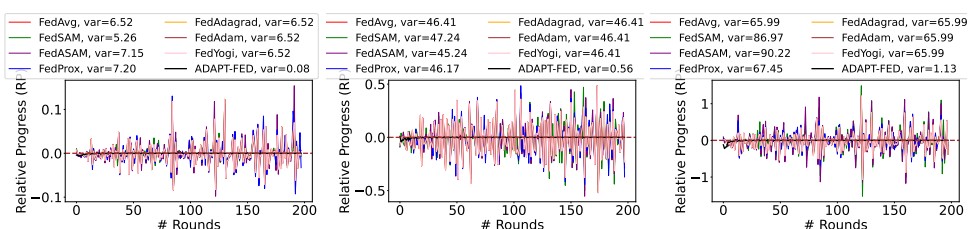

Figure 12: Stability of the training loss of ADAPT-FED and baseline algorithms on 10 clients (CIFAR100 noniid-ness $\alpha = 0.3$, $\eta_0 = 0.1$). ADAPT-FED exhibits more stable convergence compared to baselines.

## F.1 OVERVIEW OF INSTABILITY-AWARE $\beta$ SCHEDULES

Let $\boldsymbol{RP}^k = \{RP_1^k, \ldots, RP_N^k\}$ denote the observed relative progress or sharpness proxy values for client $k$ across the last $N$ training rounds. Each $\beta^k$ variant below uses a transformation of $\boldsymbol{RP}^k$ to adjust the local learning rate $\eta_t^k = \eta_0 \cdot f(\boldsymbol{RP}^k)$.

### F.1.1 SOFTMAX-BASED $\beta$

This schedule encourages more exploration when RP values are sharp by giving more weight to flatter (lower RP) regions:

$$\beta^k = \sum_{i=1}^{N} \frac{\exp(-RP_i^k)}{\sum_{j=1}^{N} \exp(-RP_j^k)}. \tag{23}$$

This corresponds to a soft attention mechanism over past instability, encouraging smoother directions.

### F.1.2 SELF-NORMALIZED $\beta$

This approach normalizes the instability magnitude:

$$\beta^k = \frac{1}{||\boldsymbol{RP}^k||_2 + \epsilon}. \tag{24}$$

This ensures scale-invariant adjustment and guards against sudden spikes in sharpness.

## F.2 COMPARATIVE ADVANTAGES

- **Softmax-based** $\beta$: Smoothly prioritizes flatter directions, especially useful when recent RP values vary dramatically.
- **Self-normalized** $\beta$: Scale-invariant and robust to overall instability magnitude.

## F.3 EMPIRICAL EVALUATION

We present ablation results comparing these variants to the $RP$-based $\beta$ scheduling mechanism. Results show that in highly unstable or $DP$-noisy settings, entropy-based and harmonic mean-based schedulers maintain more stable learning while achieving competitive generalization performance.

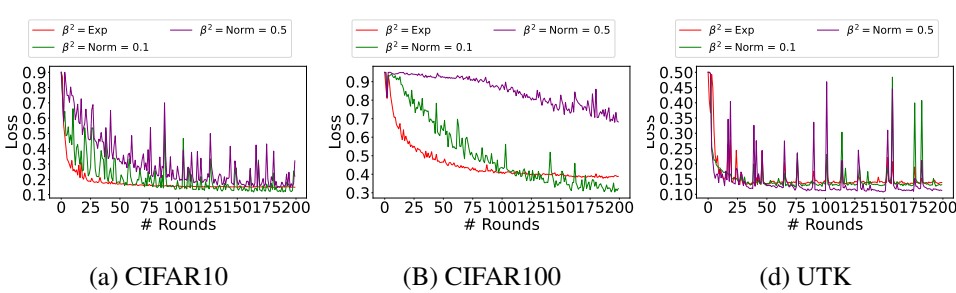

(a) CIFAR10      (B) CIFAR100      (d) UTK

Figure 13: Test loss trajectories under three $\beta$ scheduling strategies: the baseline exponential decay (Exp), and two self-normalized variants defined as $\beta = \frac{0.1}{||\boldsymbol{RP}^k||_2+\epsilon}$ and $\beta = \frac{0.5}{||\boldsymbol{RP}^k||_2+\epsilon}$, across 200 communication rounds for CIFAR-10, CIFAR-100, and UTK datasets with DP noise $\sigma^2 = 0.03$. Exp baseline $\beta$ schedules consistently lead to smoother convergence loss.

