# OpenReview forum: "ADAPT-FED: Adaptive Federated Optimization with Learning Stability"
_ICLR.cc/2026/Conference — ICLR 2026 Conference Withdrawn Submission_

### Official Review · Reviewer_hAnr · 2025-10-19

**Soundness:** 3
**Presentation:** 2
**Contribution:** 2
**Rating:** 4
**Confidence:** 3

**Summary:**

This paper aims to enhance the generalization capability of FL by tackling the unstable training issue caused by heterogeneous local data distributions. The authors propose a novel framework named ADAPT-FED, which dynamically adjusts the learning rate based on historical training instability. They perform extensive experiments, and the results demonstrate that ADAPT-FED achieves both superior generalization and convergence performance.

**Strengths:**

ADAPT-FED demonstrates significantly enhanced empirical performance across multiple benchmarks. Besides, the proposed method is effective in privacy-constrained settings.

**Weaknesses:**

1. **Limited novelty** – The proposed metric *Relative Progress (RP)* appears to use the difference in loss values across rounds, combined with DP and SAM to adjust the learning rate. The underlying idea is fairly conventional.
2. **Theoretical gap** – There is a logical jump from demonstrating training-loss descent to claiming improved generalization behavior. The link between optimization performance and generalization is not rigorously established in the current theory.
3. **Restrictive assumption** – The use of bounded-gradient assumptions is strong and may not hold in realistic federated learning settings.

**Questions:**

1. Please provide a definition of the local empirical risk $F_k$ in Eq. (1).
2. Regarding Assumption 2: please clarify the meaning of $\xi_e$ (in the expectation) and $\xi_k$ (in the gradient), respectively.
3. In federated optimization, the assumption of bounded gradients is generally considered unrealistic; it is more common to assume bounded variance of stochastic gradients when conducting convergence analysis.
4. Concerning Theorem 3.1: the authors appear to argue for generalization via the empirical-risk descent behavior, but this presents a logical jump — in most works the argument for generalization is via a theoretical upper bound on the generalization error. Additionally, from the PAC-Bayes/information-theoretic generalization viewpoint, the variance of additive Gaussian noise on gradients appears in the denominator of the generalization bound, which seems at odds with the manuscript’s claim that such noise *worsens* generalization.
5. Proposition 1 reads more like a *definition*. What exactly is the relationship between the proof in Eq. (4) and this “proposition”?
6. The idea of leveraging the difference in loss values between successive rounds to adjust the learning rate seems not particularly novel. Is the novelty here primarily coming from the combination with DP + SAM?
7. Regarding Theorem 5.1: in non-convex optimization one typically presents a convergence rate bound in terms of the minimum (over the first $T$ rounds) gradient norm. Furthermore, note that the $\bar{RP}$ term appears directly in the upper bound — the manuscript should provide more explicit derivation of how the scale/order of $\bar{RP}$ affects the convergence rate for the reader to interpret.
8. Concerning the experimental section:
   - In line 362 the text says “benchmarks datasets CIFAR10, CIFAR10, and UTK,” which lists CIFAR-10 twice.
   - In line 374 the phrase “their impact on model generalization and convergence speed are reported in ??” appears — please clarify what “??” refers to.
9. It is recommended to consider a feature-skew scenario in the experiments to better verify the robustness of the proposed method. Also, the number of clients is set to just 10 and 20, which is rather small — this may not adequately demonstrate the proposed algorithm’s robustness under heavy heterogeneity. Besides, the mean values computed by multi-runs should be reported with its corresponding standard error.

---

### Official Review · Reviewer_jhWW · 2025-10-27

**Soundness:** 1
**Presentation:** 2
**Contribution:** 2
**Rating:** 2
**Confidence:** 4

**Summary:**

ADAPT-FED introduces an adaptive federated learning framework that enhances stability and generalization under non-IID data and differential privacy (DP) noise. It proposes Relative Progress (RP) as a real-time metric for training instability and dynamically adjusts local learning rates based on its moving average. Built on FedSAM’s sharpness-aware optimization, ADAPT-FED increases learning rates during stable periods and reduces them when instability is detected. A timeout-abandonment mechanism mitigates straggler effects, while a fully distributed protocol adapts to device availability. Theoretical convergence guarantees are provided under standard assumptions. Experiments on CIFAR-10, CIFAR-100, and UTK datasets show ADAPT-FED significantly improves training efficiency, convergence robustness, and model accuracy.

**Strengths:**

ADAPT-FED proposes a lightweight “Relative Progress” cue that allows clients to autonomously scale the learning rate under non-IID and DP noise. Across CIFAR-10/100 and UTK, it yields higher accuracy, a flatter Hessian, lower loss variance, and faster convergence.

**Weaknesses:**

1. RP lacks theory that links its magnitude to curvature or generalisation, so the scheduler’s behaviour on new landscapes or models remains empirical.
2. RP is computed from noisy clipped gradients; its variance under DP is not derived, so the rule may chase noise instead of true progress.
3. All experiments use VGG/ResNet-18 on CIFAR data with 10–20 clients; no evidence on language, transformer, or large-scale tasks is provided.
4. The citation “are reported in ??.” on line 374 appears to be incomplete or improperly formatted. It seems a reference to an appendix or section was intended but not correctly inserted. Please update this to clearly indicate where the detailed ablation studies are reported

**Questions:**

There are some questions as follows:

1. What theoretical evidence supports the use of Relative Progress (RP) as a reliable indicator of training stability in federated learning? How does the paper ensure that RP remains valid across varying model architectures?

2. Since RP is computed using noisy gradients and loss differences under differential privacy, how does the paper account for the corruption of RP estimates by DP noise? Is there a robust estimator or correction mechanism in place to ensure that the adaptive learning rate remains reliable in noisy settings?

3. The adaptive learning rate schedule depends heavily on hyperparameters. What theoretical or automated guidance does the paper provide for selecting hyperparameters?

4. The experiments are limited to image classification tasks using VGG-style models on CIFAR-10, CIFAR-100, and UTK. Has the method been evaluated on other types of models (e.g., Transformers, RNNs) or domains (e.g., NLP, time series)?

5. The baseline comparisons primarily include older federated learning methods. Why does the paper not compare against more recent state-of-the-art approaches from 2024–2025?

---

### Official Review · Reviewer_Arrn · 2025-10-30

**Soundness:** 3
**Presentation:** 1
**Contribution:** 2
**Rating:** 4
**Confidence:** 4

**Summary:**

This paper proposes ADAPT-FED, a federated learning (FL) framework that adaptively regulates the learning rate based on Relative Progress (RP) — a metric designed to quantify training stability.
The method aims to improve both generalization and convergence in heterogeneous federated settings, especially when incorporating differential privacy (DP) noise.
ADAPT-FED monitors the variance of RP across training rounds and dynamically adjusts local learning rates:Increasing the rate when training is stable.Decreasing it when instability arises.

**Strengths:**

The paper directly tackles the problem of training instability caused by client heterogeneity and DP noise — an underexplored but crucial issue for generalization in real-world FL deployments.

Using RP as an indicator for training stability is a creative and theoretically grounded idea that bridges optimization dynamics with generalization control.

**Weaknesses:**

The paper primarily compares against classical methods (FedAvg, FedProx, FedAdam, FedYogi) and earlier sharpness-aware approaches (FedSAM, FedASAM)，Lack of comparison with the latest federated learning methods.

Since ADAPT-FED adjusts learning rates and may influence the number of required rounds, communication cost could change significantly.However, the paper does not analyze total communication time, bandwidth requirements, or update frequency, which are critical metrics in FL performance evaluation.


While RP is intuitively linked to stability, the paper does not rigorously prove or bound the relationship between RP variation and expected generalization error.The connection remains largely empirical, reducing theoretical completeness.


There are some citations errors.

**Questions:**

Can the authors provide communication cost analysis, e.g., number of rounds to convergence vs. total transmitted bytes?
Will the computation of RP significantly increase the time complexity?
Please add comparisons with the latest federated learning related work.
The method is currently combined with the sam optimizer. It is recommended to add result comparisons using sgd or adam optimizers.
It would be helpful to add a theoretical sketch showing how RP variance bounds the generalization gap.
It is recommended to optimize the paper's format.

---

### Official Review · Reviewer_bS3F · 2025-11-03

**Soundness:** 3
**Presentation:** 3
**Contribution:** 2
**Rating:** 4
**Confidence:** 3

**Summary:**

This paper introduces ADAPT-FED, a federated learning framework designed to address instability and poor generalization arising from heterogeneous data and differentially private training. The core idea is to dynamically regulate each client’s local learning rate according to a novel relative progress (RP) metric, which quantifies training stability. The paper provides both theoretical guarantees and extensive empirical evaluations, demonstrating improved stability, convergence, and generalization on standard federated benchmarks (CIFAR10, CIFAR100, and UTK) compared to SOTA methods.

**Strengths:**

- The paper addresses a central and impactful challenge in federated learning: the tension between training instability and generalization, especially under heterogeneity and privacy constraints.
- ADAPT-FED’s adaptive learning rate schedule, guided by the relative progress (RP) metric integrated into the optimization process, is a conceptually interesting and potentially useful contribution. This is more direct than prior flatness-focused or sharpness-aware minimization adaptations.
- The theoretical analysis provides explicit, nontrivial upper bounds on convergence and stability, grounded in reasonable assumptions and extending to the differential privacy regime. For instance, Theorem 5.1 delivers a client-wise adaptive bound that ties RP to progress and error terms.
- The empirical evaluation is broad, covering CIFAR10, CIFAR100, and UTK datasets, with ablations on hyperparameters and comparisons against a diverse range of FL baselines (FedAvg, FedSAM, FedASAM, FedProx, FedYogi, etc.).

**Weaknesses:**

-  While the paper discusses the RP metric and its theoretical significance in depth, several symbols and notations in the central equations—such as those in Theorem 3.1, Proposition 1, and the RP formula itself—lack clear, standalone definitions in the main text. For example, variables like $\bar{g}_k$, $t^s$, and several others appear suddenly. The dependence of some steps on DP noise is described informally. This is an obstacle to both theoretical transparency and implementation.
- Algorithm 1, while comprehensive, leaves open questions about certain practical choices (e.g., use of exponential averaging in RP, handling of negative RP before exponentiation, choice/rationale for $\beta = 2$, etc.). The explanations motivating some hyperparameter choices (e.g., why specific grid choices for $\rho$, $\mu$) are thin, and the practical guidance is quite minimal for generalization to other tasks.
- Although the experiments are diverse, there is a distinct focus on three FL benchmarks with relatively moderate non-IID severity and standard architectures. The evaluation does not convincingly probe the method’s limits with more adversarial heterogeneity, extreme client dropout, or strongly non-standard data splits, which would be essential for general FL deployment. There is no reported evaluation on real-world or larger-scale horizontal or vertical FL datasets.
-  There is little granularity in the description of how the SOTA baselines (especially adaptive client-side or bias-mitigated methods) are implemented or tuned. For instance, whether FedDANE or AdaBest (should they be included) could be tuned for improved results compared to those in the present baselines is not considered.
- In several formulas (e.g., the clipping and DP noise equations, Equation 14 in Algorithm 1), the variables (e.g., $L_l$ in the noise term) are not defined anywhere in the section. These cause confusion.

**Questions:**

Please refer to Weaknesses

---

### Note · Authors · 2025-11-28

I have read and agree with the venue's withdrawal policy on behalf of myself and my co-authors.